# Fair Bayesian Model-Based Clustering

## Abstract

Fair clustering has become a socially significant task with the advancement of machine learning and the growing demand for trustworthy AI. Group fairness ensures that the proportions of each sensitive group are similar in all clusters. Most existing fair clustering methods are based on the $K$-means clustering and thus require the distance between instances and the number of clusters to be given in advance. To resolve this limitation, we propose a fair Bayesian model-based clustering called Fair Bayesian Clustering (FBC). We develop a specially designed prior which puts its mass only on fair clusters, and implement an efficient MCMC algorithm. The main advantage of FBC is its flexibility in the sense that it can infer the number of clusters, can process data where the choice of a reasonable distance is difficult (e.g., categorical data), and can reflect a constraint on the sizes of each cluster. We illustrate these advantages by analyzing real-world datasets.

## 1 Introduction

With the rapid development of machine learning-based technologies, algorithmic fairness has been considered as an important social consideration when making machine learning-based decisions. Among diverse tasks combined with algorithmic fairness, fair clustering (Chierichetti et al., 2017) has received much interest, which ensures that clusters maintain demographic fairness across sensitive attributes such as gender or race. However, most of existing fair clustering methods can be seen as modifications of $K$-means clustering algorithms, and thus they require (i) the number of clusters to be fixed and (ii) the distance between two instances given a priori, limiting their adaptability to various kinds of real-world datasets, where the optimal number of clusters is unknown and/or it is hard to define a reasonable distance (e.g., categorical data).

Among standard (fairness-agnostic) clustering approaches, various Bayesian models that can infer the number of clusters have been proposed including mixture models with unknown components (Richardson & Green, 1997; Nobile & Fearnside, 2007; McCullagh & Yang, 2008; Miller & Harrison, 2018) and mixture of Dirichlet process models (Ferguson, 1973; Antoniak, 1974; Escobar & West, 1995; Neal, 2000). For fair Bayesian mixture models, we first modify the standard Bayesian mixture model so that a prior concentrates on fair mixture models (called the fair prior) and so does the posterior distribution. In general, however, computation of the posterior on a constraint parameter space would be computationally demanding and frequently infeasible if the constraint is not designed carefully (Brubaker et al., 2012; Sen et al., 2018; Duan et al., 2020).

To address such limitations of existing fair clustering methods and to leverage the flexibility of Bayesian model-based approaches, we aim to develop a fair clustering algorithm based on a Bayesian framework. In Bayesian frameworks, the goal is to explore high-posterior regions of the parameter space. In the context of fair clustering, however, the posterior should be explored only within the fair region. In turn, this requires that the prior mass to be concentrated on the fair space, rather than on the entire unconstrained space. Motivated by this, we develop a fair prior based on the idea of matching instances from different sensitive groups, which has already been successfully implemented for fair supervised learning (Kim et al., 2025a) and non-Bayesian fair clustering (Chierichetti et al., 2017; Kim et al., 2025b), but this is the first attempt for Bayesian fair clustering. An important advantage is that our proposed fair prior does not involve any explicit constraint on the parameters and thus posterior approximation by an MCMC algorithm can be done without much difficulty. Our matching-based formulation in Section 2 is, to our knowledge, the first to explicitly construct such a prior restricted to the fair space.

A further benefit of our proposed algorithm is that it can be applied to both cases when the number of clusters is known (fixed) and unknown. To infer the unknown number of clusters, we treat the number of clusters as a parameter to be inferred. Specifically, we propose a fair Bayesian mixture model and develop an MCMC algorithm to approximate the posterior distribution of the number of clusters as well as the cluster centers and cluster assignments. Figure 1 presents an example where the choice of the number of clusters under the fairness constraint a priori would be difficult in practice.

Beyond the inference of the number of clusters, our proposed algorithm has further benefits: (i) it can be applied to various data types, since it is a model-based clustering method, and (ii) it can perform clustering under the constraint that the cluster sizes must be upper-bounded, since it is a Bayesian clustering.

Our main contributions are summarized as follows:

⋄ We propose a definition of the fair mixture model and develop a novel MCMC algorithm to approximate the posterior distribution of fair mixture models, called Fair Bayesian Clustering (FBC).
⋄ Experimentally, we show that FBC (i) is competitive to existing non-Bayesian fair clustering methods when the number of clusters is given, (ii) can infer a proper number of clusters well when the number of clusters is unknown, (iii) can use both continuous and categorical data, and (iv) can infer fair clusters under certain constraints on the sizes of each cluster.

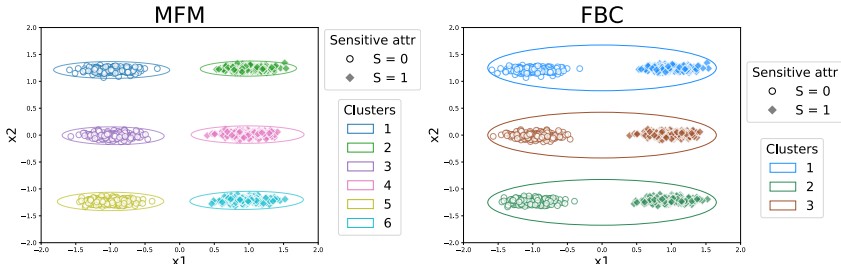

Figure 1: Toy example comparing two Bayesian clustering methods with and without considering fairness. MFM (Mixture of Finite Mixtures (Miller & Harrison, 2018)) is a standard Bayesian method that infers $K$. Data are synthetically generated from the 6-component Gaussian mixture dataset (see Section E.1 for details), and MFM infers $K = 6$, whereas the optimal number of clusters obtained by FBC is $K = 3$. This example indicates that the choice of $K$ by use of fairness-agnostic clustering algorithms would be misleading for fair clustering.

## 1.1 RELATED WORKS

**Fair clustering**   Given a pre-specified sensitive attribute (e.g., gender or race), the concept of fair clustering is first introduced by Chierichetti et al. (2017), with the aim of ensuring that the proportion of each sensitive group within each cluster matches the overall proportion in the entire dataset. This fairness criterion is commonly referred to as group fairness, and is also known as proportional fairness. Recently, several algorithms have been developed to maximize clustering utility under fairness constraints: Chierichetti et al. (2017); Backurs et al. (2019) transform training data to fair representations to achieve fairness guarantee prior to clustering, Kleindessner et al. (2019); Ziko et al. (2021); Li et al. (2020); Zeng et al. (2023) incorporate fairness penalties directly during the clustering process, and Bera et al. (2019); Harb & Lam (2020) refine cluster assignments with fixed pre-determined cluster centers. These methods, however, require the number of clusters to be given in advance. This limitation, i.e., the lack of fair clustering algorithms that are adaptive to the unknown number of clusters, serves as the motivation for this study. In addition, model-based clustering can be applied to data for which a meaningful distance is difficult to define (e.g., categorical data).

**Bayesian model-based clustering**   The mixture model is a model-based clustering approach, where each instance (or observation) is assumed to follow a mixture of parametric distributions independently (Ouyang et al., 2004; Reynolds et al., 2009). Popular examples of parametric distributions are the Gaussian (Maugis et al., 2009; Yang et al., 2012; Zhang et al., 2021), the student-t

(Peel & McLachlan, 2000), the skew-normal (Lin et al., 2007), and the categorical (Pan & Huang, 2014; McLachlan & Peel, 2000).

Bayesian approaches have been popularly used for inference of model-based clustering since they can infer the number of clusters as well as cluster centers. There are several well-defined Bayesian model-based clustering with unknown number of clusters including Mixture of Finite Mixtures (MFM) (Richardson & Green, 1997; Nobile & Fearnside, 2007; McCullagh & Yang, 2008; Miller & Harrison, 2018) and Dirichlet Process Mixture (DPM) (Ferguson, 1973; Antoniak, 1974; Escobar & West, 1995; Neal, 2000). There also exist various MCMC algorithms for MFM and DPM, such as Reversible Jump Markov Chain Monte Carlo (RJMCMC) (Richardson & Green, 1997) and Jain-Neal split-merge algorithm (Jain & Neal, 2004).

## 2 FAIR MIXTURE MODEL FOR CLUSTERING

We consider group fairness, as it is one of the most widely studied notions of fairness (Chierichetti et al., 2017; Backurs et al., 2019; Ziko et al., 2021; Li et al., 2020; Zeng et al., 2023; Bera et al., 2019; Harb & Lam, 2020; Kim et al., 2025b). For simplicity, we only consider a binary sensitive attribute, but provide a method of treating a multinary sensitive attributes in Section C. Let $s \in \{0,1\}$ be a binary sensitive attribute known a priori. We define two sets of instances (observed data) from the two sensitive groups as $\mathcal{D}^{(s)} = \{X_i^{(s)} : X_i^{(s)} \in \mathcal{X} \subseteq \mathbb{R}^d\}_{i=1}^{n_s}$ for $s \in \{0,1\}$, where $\mathcal{X}$ is the support of $X$, $n_s$ is the number of instances in the sensitive group $s$, and $d$ is the number of features. Let $\mathcal{D} := \mathcal{D}^{(0)} \cup \mathcal{D}^{(1)}$ be the set of the entire instances.

**Standard mixture model** The standard finite mixture model (Ouyang et al., 2004; Reynolds et al., 2009; Maugis et al., 2009; Yang et al., 2012; Zhang et al., 2021) without fairness is given as

$$X_1, \ldots, X_n \overset{\text{i.i.d.}}{\sim} \sum_{k=1}^K \pi_k f(\cdot|\theta_k) \tag{1}$$

where $n := n_0 + n_1$ and $X_i := X_i^{(0)}$ for $i \in \{1, \ldots, n_0\}$ and $X_{i-n_0}^{(1)}$ for $i \in \{n_0+1, \ldots, n_0+n_1\}$. In view of clustering, $K$ is considered to be the number of clusters, $\boldsymbol{\pi} := (\pi_k)_{k=1}^K \in \mathcal{S}_K$ (the $K$-dimensional simplex) are the proportions of instances belonging to the $k^{\text{th}}$ cluster, and $f(\cdot|\theta_k)$ is the density of instances in the $k^{\text{th}}$ cluster.

An equivalent representation of the above model can be made by introducing latent variables $Z_i \in [K], i \in [n]$ as the following. Note that the latent variable $Z_i$ takes the role of the cluster assignment of $X_i$ for all $i \in [n]$. That is, when $Z_i = k$, we say that $X_i$ belongs to the $k^{\text{th}}$ cluster.

$$Z_1, \ldots, Z_n \overset{\text{i.i.d.}}{\sim} \text{Categorical}(\boldsymbol{\pi}) \tag{2}$$
$$X_i|Z_i \sim f(\cdot|\theta_{Z_i}) \tag{3}$$

**Fair mixture model** To define a fair mixture model, we first generalize the formulation of the standard finite mixture model in Equations (2) and (3) by considering the dependent latent variables, which we call the *generalized finite mixture model* in this paper. Let $\boldsymbol{Z} := (Z_1, \ldots, Z_n)$. Then, the generalized finite mixture model is defined as:

$$\boldsymbol{Z} \sim G(\cdot) \tag{4}$$
$$X_i|Z_i \sim f(\cdot|\theta_{Z_i}) \tag{5}$$

where $G$, i.e., the joint distribution of $\boldsymbol{Z}$, has its support on $[K]^n$. When $G$ is equal to Categorical$(\boldsymbol{\pi})^n$ (i.e., $Z_i, \forall i \in [n]$ independently follows Categorical$(\boldsymbol{\pi})$), the generalized mixture model becomes the standard finite mixture model in Equations (2) and (3).

We also define the (group) fairness level of $\boldsymbol{Z}$:

$$\Delta(\boldsymbol{Z}) := \frac{1}{2} \sum_{k=1}^K \left| \sum_{i=1}^{n_0} \mathbb{I}(Z_i^{(0)} = k)/n_0 - \sum_{j=1}^{n_1} \mathbb{I}(Z_j^{(1)} = k)/n_1 \right| \in [0,1] \tag{6}$$

where $Z_i^{(0)} = Z_i$ for $i \in \{1, \ldots, n_0\}$ and $Z_j^{(1)} = Z_{j+n_0}$ for $j \in \{1, \ldots, n_1\}$. We say that $\mathbf{Z}$ *is fair with fairness level $\epsilon$* if $\mathbf{Z} \in \mathcal{Z}_\epsilon^{\text{Fair}}$, where $\mathcal{Z}_\epsilon^{\text{Fair}} := \{\mathbf{Z} : \Delta(\mathbf{Z}) \leq \epsilon\}$. In turn, for a given generalized mixture model, if the support $G$ is confined on $\mathcal{Z}_\epsilon^{\text{Fair}}$, we say it is fair with level $\epsilon$. Note that the notion of $\Delta$ has been already implemented in previous works (Kim et al., 2025b). See Section A.4 for more details.

## 2.1 CHOICE OF $G$ FOR PERFECT FAIRNESS: $\Delta(\mathbf{Z}) = 0$

The art of Bayesian analysis of the fair mixture model is to parameterize $G$ such that posterior inference becomes computationally feasible. For example, we could consider Categorical$(\boldsymbol{\pi})^n$ (the distribution of independent $Z_i$) restricted to $\mathcal{Z}_0^{\text{Fair}}$ as a candidate for $G$. While developing an MCMC algorithm for this distribution would not be impossible, but it would be quite challenging.

In this section, we propose a novel distribution for $G$ in the perfectly fair (i.e., $\Delta(\mathbf{Z}) = 0$) mixture model with which a practical MCMC algorithm for posterior inference can be implemented. To explain the main idea of our proposed $G$, we first consider the simplest case of balanced data where $n_0 = n_1$ in Section 2.1.1. We then discuss how to handle the case of $n_0 \neq n_1$ in Section 2.1.2.

### 2.1.1 CASE OF $n_0 = n_1$

Let $\bar{n} := n_0 = n_1$. A key observation is that any fair $\mathbf{Z}$ corresponds to a matching map between $[\bar{n}]$ and $[\bar{n}]$ (a permutation on $[\bar{n}]$), which is stated in Proposition 2.1 with its proof in Section A. Figure 2 illustrates the relationship between a fair $\mathbf{Z}$ and a matching map $\mathbf{T} : [\bar{n}] \to [\bar{n}]$.

**Proposition 2.1.** $\mathbf{Z} \in \mathcal{Z}_0^{\text{Fair}} \iff$ *There exists a matching map $\mathbf{T}$ such that $Z_j^{(1)} = Z_{\mathbf{T}(j)}^{(0)}, \forall j \in [\bar{n}]$.*

We utilize the above proposition to define a fair distribution $G$. The main idea is that $G$ *assigns two matched data to a same cluster once a matching map is given*. To be more specific, the proposed distribution is parametrized by $\boldsymbol{\pi}$ and $\mathbf{T}$, which is defined by $Z_1^{(0)}, \ldots, Z_{\bar{n}}^{(0)} \overset{\text{i.i.d}}{\sim}$ Categorical$(\boldsymbol{\pi})$ and $Z_j^{(1)} = Z_{\mathbf{T}(j)}^{(0)}$. It is easy to see that $G$ is perfectly fair. We write $G(\cdot|\boldsymbol{\pi}, \mathbf{T})$ for such a distribution.

Denote $\mathcal{Z}^{\text{Fair}}(\mathbf{T})$ as the support of $G(\cdot|\boldsymbol{\pi}, \mathbf{T})$. Note that $\mathcal{Z}^{\text{Fair}}(\mathbf{T}) \subset \mathcal{Z}_0^{\text{Fair}}$, so one might worry that the support of $G(\cdot|\boldsymbol{\pi}, \mathbf{T})$ is too small. Proposition 2.1, however, implies that $\cup_{\mathbf{T} \in \mathcal{T}} \mathcal{Z}^{\text{Fair}}(\mathbf{T}) = \mathcal{Z}_0^{\text{Fair}}$, where $\mathcal{T}$ is the set of all matching maps, and thus we can put a prior mass on $\mathcal{Z}_0^{\text{Fair}}$ by putting a prior on $\mathbf{T}$ (as well as $\boldsymbol{\pi}$) accordingly.

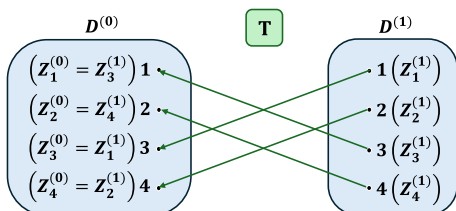

Figure 2: An example of fair $\mathbf{Z}$ when $\bar{n} = 4$, where $\mathbf{T}$ is given as $(\mathbf{T}(1), \mathbf{T}(2), \mathbf{T}(3), \mathbf{T}(4)) = (3, 4, 1, 2)$.

A crucial benefit of the proposed $G$ is that $\boldsymbol{\pi}$ and $\mathbf{T}$ are not intertwined, hence they can be selected independently. As a result, we can find a fair mixture model without any additional parameter constraints. It is noteworthy that, conditional on $\mathbf{T}$, the fair mixture model can be written similarly to the standard mixture model in Equations (4) and (5) as :

$$(Z_1^{(0)}, \ldots, Z_{\bar{n}}^{(0)}) \sim \text{Categorical}(\boldsymbol{\pi})^{\bar{n}} \tag{7}$$

$$X_{\mathbf{T}(j)}^{(0)}, X_j^{(1)}|Z_{\mathbf{T}(j)}^{(0)} \overset{\text{ind}}{\sim} f(\cdot|\theta_{Z_{\mathbf{T}(j)}^{(0)}}) \tag{8}$$

### 2.1.2 CASE OF $n_0 \neq n_1$

Without loss of generality, we assume that $n_0 < n_1$ such that $n_1 = \beta n_0 + r$ for nonnegative integers $\beta$ and $r < n_0$. We first consider the case of $r = 0$ because we can modify $G$ for the balanced case easily. When $r \neq 0$, the situation is complicated since there is no one-to-one correspondence between fairness and matching map, and thus we propose a heuristic modification.

**Case of $r = 0$ :** A function $\mathbf{T}$ from $[n_1]$ to $[n_0]$ is called a *matching map* if (i) it is onto and (ii) $|\mathbf{T}^{-1}(i)| = \beta$ for all $i \in [n_0]$. Let $\mathcal{T}$ be the set of all matching maps. Then, we have Proposition 2.2,

which is similar to Proposition 2.1 for the balanced case. See Section A for its proof. Note that for this case, $\mathbf{T}$ can be seen as a map to build fairlets.

**Proposition 2.2.** $\boldsymbol{Z} \in \mathcal{Z}_0^{\text{Fair}} \iff$ *There exists a matching map* $\mathbf{T}$ *s.t.* $Z_j^{(1)} = Z_{\mathbf{T}(j)}^{(0)}, \forall j \in [n_1]$.

For given $\boldsymbol{\pi} \in \mathcal{S}_K$ and $\mathbf{T} \in \mathcal{T}$, we define a fair distribution $G(\cdot|\boldsymbol{\pi}, \mathbf{T})$ as $Z_1^{(0)}, \ldots, Z_{n_0}^{(0)} \overset{\text{i.i.d}}{\sim}$ Categorical$(\boldsymbol{\pi})$ and $Z_j^{(1)} = Z_{\mathbf{T}(j)}^{(0)}$. Similar to the balanced case, we have $\cup_{\mathbf{T} \in \mathcal{T}} \mathcal{Z}^{\text{Fair}}(\mathbf{T}) = \mathcal{Z}_0^{\text{Fair}}$.

**Case of** $r > 0$ **:** It can be shown (see Proposition A.1 in Section A for the proof) that for a given fair $\boldsymbol{Z} \in \mathcal{Z}_0^{\text{Fair}}$, there exists a function $\mathbf{T}$ from $[n_1]$ to $[n_0]$ such that it is onto, $|\mathbf{T}^{-1}(i)|$ is either $\beta$ or $\beta + 1$ and $|R_{\mathbf{T}}| = r$, where $R_{\mathbf{T}} = \{i : |\mathbf{T}^{-1}(i)| = \beta + 1\}$. Let $\mathcal{T}$ be the set of all such matching maps (functions satisfying these conditions).

A difficulty arises since the converse is not always true. To resolve this difficulty, we propose a heuristic modification of the definition of 'fairness of $\boldsymbol{Z}$'. Let $R$ be a subset of $[n_0]$ with $|R| = r$, and let $\mathcal{T}_R$ be a subset of $\mathcal{T}$ such that $R_{\mathbf{T}} = R$. Then, we say that $\boldsymbol{Z}$ is fair if there exists $\mathbf{T} \in \mathcal{T}_R$ such that $Z_j^{(1)} = Z_{\mathbf{T}(j)}^{(0)}$. Note that a fair $\boldsymbol{Z}$ may not belong to $\mathcal{Z}_0^{\text{Fair}}$ but the violation of fairness would be small when $|C_k^{(0)}|/n_0 \approx |C_k^{(0)} \cap R|/r$. A theoretical upper bound on the fairness violation ($\Delta(\boldsymbol{Z})$) is given in Proposition A.3, and it becomes zero when $|C_k^{(0)}|/n_0 = |C_k^{(0)} \cap R|/r$ as shown in Proposition A.2.

In this paper, we consider two feasible candidates for $R$: (i) a random subset of $[n_0]$ and (ii) the index of samples closest to the cluster centers obtained by a certain clustering algorithm to $\mathcal{D}^{(0)}$ with $K = r$. Finally, we can define $G(\cdot|\boldsymbol{\pi}, \mathbf{T})$ for $\boldsymbol{\pi} \in \mathcal{S}_K$ and $\mathbf{T} \in \mathcal{T}_R$ similarly to that for the balanced case. See Section A.1 for the detailed discussion and the motivation of the two candidates. Our numerical studies in Section 5.4 confirm that the two proposed choices of $R$ work quite well.

## 2.2 Choice of $G$ for non-perfect fairness: $\Delta(\boldsymbol{Z}) > 0$

Once $\boldsymbol{\pi} \in \mathcal{S}_K$ and $\mathbf{T} \in \mathcal{T}_R$ are given, we can modify $G(\cdot|\boldsymbol{\pi}, \mathbf{T})$ to have a distribution whose support is included in $\mathcal{Z}_\varepsilon^{\text{Fair}}$. The main idea of the proposed modification is to select $m$ many samples from $\mathcal{D}^{(1)}$ and to assign independent clustering labels to them instead of matching them to the corresponding samples in $\mathcal{D}^{(0)}$. Let $E$ be a subset of $[n_1]$ with $|E| = m$. We consider a latent variable $\mathbf{T}_0 : [n_1] \to [n_0]$ that is an arbitrary function from $[n_1]$ to $[n_0]$. Then, we let $Z_j^{(1)} = Z_{\mathbf{T}(j)}^{(0)}$ for $j \in [n_1] \setminus E$ and $Z_j^{(1)} = Z_{\mathbf{T}_0(j)}^{(0)}$ for $j \in E$. In other words, $\mathcal{D}^{(1)}$ is masked by $E$ : (i) the masked data in $E$ are assigned by $\mathbf{T}_0$, (ii) while the unmasked data in $[n_1] \setminus E$ are still assigned by $\mathbf{T}$. It can be shown that the support of the distribution $G(\cdot|\boldsymbol{\pi}, \mathbf{T}, \mathbf{T}_0, E)$ belongs to $\mathcal{Z}_\varepsilon^{\text{Fair}}$ with $\varepsilon = m/n_1$, provided that $r = 0$. See Proposition A.4 in Section A for the theoretical proof.

## 3 Fair Bayesian modeling

We first propose a fair mixture model based on Equations (7) and (8) as:

$$\boldsymbol{\pi}|K \sim \text{Dirichlet}_K(\gamma, \ldots, \gamma), \gamma > 0 \tag{9}$$

$$Z_i^{(0)}|\boldsymbol{\pi} \overset{\text{i.i.d.}}{\sim} \text{Categorical}(\cdot|\boldsymbol{\pi}), \quad \forall i \in [n_0] \tag{10}$$

$$Z_j^{(1)} = \begin{cases} Z_{\mathbf{T}(j)}^{(0)}, & \forall j \in [n_1] \setminus E \\ Z_{\mathbf{T}_0(j)}^{(0)}, & \forall j \in E \end{cases} \tag{11}$$

$$X_i^{(0)}|Z_i^{(0)} \sim f(\cdot|\theta_{Z_i^{(0)}}), X_j^{(1)}|Z_j^{(1)} \sim f(\cdot|\theta_{Z_j^{(1)}}) \tag{12}$$

In this model, $\boldsymbol{\theta} = (\theta_1, \theta_2, \ldots), E, \mathbf{T}$ and $\mathbf{T}_0$ as well as the number of clusters $K$ are the parameters to be inferred. Aprior, we assume that each parameter is independent: $K \sim p_K(\cdot), E \sim p_E(\cdot), \mathbf{T} \sim p_{\mathbf{T}}(\cdot), \mathbf{T}_0 \sim p_{\mathbf{T}_0}(\cdot)$, and $\theta_1, \theta_2 \ldots, \overset{\text{ind}}{\sim} H$.

**Prior for** $K$    For $K$, we consider $K \sim p_K(\cdot)$, where $p_K$ is a probability mass function on $\{1, 2, \dots\}$. In this study, we use Geometric$(\kappa), \kappa \in (0, 1)$ for $p_K$ following (Miller & Harrison, 2018), where $\kappa$ is a hyperparameter. Further, we could consider a hierarchical prior for $\kappa$ as well to reduce the dependency of posterior to the selection of $\kappa$ (see Section D for details).

**Prior for** $\theta$    Usually, we choose a conjugate distribution of $f(\cdot|\theta)$ for $H$. For example, when $f(\cdot|\theta)$ is the density of the Gaussian distribution, and $\theta$ consists of the mean vector and diagonal covariance matrix as $\theta = (\boldsymbol{\mu}, \boldsymbol{\lambda}^{-1}) \in \mathbb{R}^d \times \mathbb{R}^d$, where $\boldsymbol{\mu} = (\mu_j)_{j=1}^d$ and $\boldsymbol{\lambda} = (\lambda_j)_{j=1}^d$. We set $\lambda_j = \lambda$ for all $j \in [d]$ where $(\mu_j, \lambda), j \in [d]$ is independent and follow $\lambda \sim \text{Gamma}(a, b)$ for some $a, b$ and $\mu_j|\lambda \sim \mathcal{N}(0, \lambda^{-1}), j \in [d]$ for $H$.

**Prior for** $\mathbf{T}, \mathbf{T}_0,$ **and** $E$    Motivated by (Volkovs & Zemel, 2012), we construct a prior of $\mathbf{T}$ with its support $\mathcal{T}_R$ based on the energy defined in Definition 3.1 below. The energy of a random matching map $\mathbf{T}$ is defined as below, which measures the similarity between two matched data. See Section E.2 for the choice of $D$ in Definition 3.1. For the prior, we let $p_{\mathbf{T}}(\mathbf{T}) \propto \mathbf{e}(\mathbf{T})\mathbb{I}(\mathbf{T} \in \mathcal{T}_R)$. For $\mathbf{T}_0$, we use the uniform distribution on $\Pi_n$. For $E$, we use the uniform distribution on $[n_1 : m]$, where $[n_1 : m]$ is the collection of all subsets of $[n_1]$ with size $m$.

**Definition 3.1** (Energy of a matching map)**.** Let $D : \mathcal{X} \times \mathcal{X} \to \mathbb{R}_+$ be a given distance and $\Pi_n := \{\mathbf{T} : [n_1] \to [n_0]\}$. Given $\mathbf{T} \in \Pi_n$, the energy of $\mathbf{T}$ is defined by $\mathbf{e}(\mathbf{T}) = \mathbf{e}(\mathbf{T}; \tau) := \exp\left(-\sum_{j=1}^{n_1} D\left(X_{\mathbf{T}(j)}^{(0)}, X_j^{(1)}\right)/n_1\tau\right)$, where $\tau > 0$ is a pre-specified temperature constant.

## 3.1 Equivalent representations

We here present the equivalent representations of Equations (9) to (12), which enable practical implementation of an MCMC inference algorithm. The proof could be done by modifying the proof of Miller & Harrison (2018), but we present the detailed proofs in Propositions B.1 to B.3 of Section B.3 for the readers' sake.

**Model**    For a given partition $\mathcal{C}$ of $[n_0]$, an equivalent generative model to the proposed fair mixture model in Equations (9) to (12) is:

$$\phi_c \overset{\text{i.i.d.}}{\sim} H, c \in \mathcal{C} \tag{13}$$

$$X_i^{(0)} \overset{\text{ind}}{\sim} f(\cdot|\phi_c) \quad i \in c \tag{14}$$

$$X_j^{(1)} \overset{\text{ind}}{\sim} \begin{cases} f(\cdot|\phi_c) & \forall j \in [n_1] \setminus E \text{ s.t. } \mathbf{T}(j) \in c \\ f(\cdot|\phi_c) & \forall j \in E \text{ s.t. } \mathbf{T}_0(j) \in c. \end{cases} \tag{15}$$

**Priors**    The prior for $\mathcal{C}$ in this equivalent representation is $p_{\mathcal{C}}(\mathcal{C}|\mathbf{T}, \mathbf{T}_0, E) = p_{\mathcal{C}}(\mathcal{C}) = V_{n_0}(t) \prod_{c \in \mathcal{C}} \gamma^{(|c|)}$, where $t = |\mathcal{C}|, V_{n_0}(t) = \sum_{k=1}^{\infty} \frac{k_{(t)}}{(\gamma k)^{(n_0)}} p_K(k), (\gamma k)^{(n_0)} = (\gamma k + n_0 - 1)!/(\gamma k - 1)!$ and $k_{(t)} = k!/(k - t)!$. See Miller & Harrison (2018) for the derivation. The prior of $(\mathbf{T}, \mathbf{T}_0, E)$ remains the same as the prior in Section 3.

When $K$ is unknown and treated as a random variable, we use the following equivalent representation of the proposed fair Bayesian mixture model, as done in Miller & Harrison (2018). When $K$ is known as $k_*$, we only need to modify $p_K$ in the prior for $\mathcal{C}$. In specific, we modify $p_K(k) = \mathbb{I}(k = k_*)$ in $V_{n_0}(t)$, and the others remain the same.

## 4 Inference algorithm: FBC

We develop an MCMC algorithm for the equivalent representations of the proposed fair Bayesian mixture model in Section 3.1. Here, we denote $\Phi$ as the mixture parameters to be sampled (i.e., $\Phi = \mathcal{C}$ when $H$ is conjugate, or $\Phi = (\mathcal{C}, \boldsymbol{\phi})$ when $H$ is non-conjugate, where $\boldsymbol{\phi} := (\phi_c : c \in \mathcal{C})$).

The posterior sampling of $(\mathbf{T}, \mathbf{T}_0, E, \Phi) \sim p(\mathbf{T}, \mathbf{T}_0, E, \Phi|\mathcal{D})$ is done by a Gibbs sampler: (i) sampling $(\mathbf{T}, \mathbf{T}_0, E) \sim p(\mathbf{T}, \mathbf{T}_0, E|\Phi, \mathcal{D})$, and (ii) sampling $\Phi \sim p(\Phi|\mathbf{T}, \mathbf{T}_0, E, \mathcal{D})$. We name the proposed MCMC inference algorithm as **Fair Bayesian Clustering (FBC)**. In the subsequent two

subsections, we explain how to sample $(\mathbf{T}, \mathbf{T}_0, E)$ and $\Phi$ from their conditional posteriors, respectively, by using a Metropolis-Hastings (MH) algorithm. We also discuss the extension of FBC for handling a multinary sensitive attribute in Section C, with experiments on a real dataset.

**STEP 1 ▷ Sampling $(\mathbf{T}, \mathbf{T}_0, E) \sim p(\mathbf{T}, \mathbf{T}_0, E|\Phi, \mathcal{D})$**

- **(Proposal)** For the proposal distribution of $(\mathbf{T}', \mathbf{T}_0', E')$ from $(\mathbf{T}, \mathbf{T}_0, E)$, we first assume that $\mathbf{T} \to \mathbf{T}'$, $\mathbf{T}_0 \to \mathbf{T}_0'$ and $E \to E'$ are independent. For the proposal of $\mathbf{T} \to \mathbf{T}'$, we randomly select two indices $i_1$ and $i_2$ from $[n_1]$, and define:

$$\mathbf{T}'(j) := \begin{cases} \mathbf{T}(j) & \text{for } j \notin [n_1] \setminus \{i_1, i_2\} \\ \mathbf{T}(i_2) & \text{for } j = i_1 \\ \mathbf{T}(i_1) & \text{for } j = i_2. \end{cases} \tag{16}$$

  We swap only two indices to guarantee $\mathbf{T}' \in \mathcal{T}_R$. For $\mathbf{T}_0 \to \mathbf{T}_0'$, we randomly select an index $j' \in [n_1]$, then set $\mathbf{T}_0'(j') = i$ where $i \sim \text{Unif}([n_0])$ and $\mathbf{T}_0'(j) = \mathbf{T}_0(j)$ for $j \neq j'$. For $E \to E'$, we randomly swap two indices, one from $E$ and the other from $[n_1] \setminus E$. See Figure 7 in Section B.2 for an illustration of the proposal.

- **(Acceptance / Rejection)** As the randomness in the proposal of $\mathbf{T}', \mathbf{T}_0'$ and $E'$ does not depend on $\mathbf{T}, \mathbf{T}_0$ and $E$, the proposal density ratio $q((\mathbf{T}', \mathbf{T}_0', E') \to (\mathbf{T}, \mathbf{T}_0, E))/q((\mathbf{T}, \mathbf{T}_0, E) \to (\mathbf{T}', \mathbf{T}_0', E'))$ is equal to 1. Hence, the acceptance probability of a proposal $(\mathbf{T}', \mathbf{T}_0', E')$ is

$$\alpha(\mathbf{T}', \mathbf{T}_0', E') = \min\left\{1, e(\mathbf{T}')\mathcal{L}(\mathcal{D}; \Phi, \mathbf{T}', \mathbf{T}_0', E')/e(\mathbf{T})\mathcal{L}(\mathcal{D}; \Phi, \mathbf{T}, \mathbf{T}_0, E)\right\}. \tag{17}$$

  See Section B.2 for the calculation details of the acceptance probability. We repeat the MH sampling of $(\mathbf{T}', \mathbf{T}_0', E')$ multiple times before sampling $\Phi$, which helps accelerate convergence. In our experiments, we perform this repetition 10 times. This additional computation is minimal, even for large datasets, since the acceptance probability requires calculating the likelihood only for instances whose assigned clusters change. The maximum number of such instances with changed clusters is 4 (2 for $\mathbf{T}'$, 1 for $\mathbf{T}_0'$ and 1 for $E'$).

**STEP 2 ▷ Sampling $\Phi \sim p(\Phi|\mathbf{T}, \mathbf{T}_0, E, \mathcal{D})$** Sampling from $p(\Phi|\mathbf{T}, \mathbf{T}_0, E, \mathcal{D})$ can be done similar to the sampling algorithm for the standard mixture model when $(\mathbf{T}, \mathbf{T}_0, E)$ is given. Specifically, we mimic the procedure of Miller & Harrison (2018), which utilizes DPM inference algorithms from Neal (2000); MacEachern & Müller (1998). See Section B.3 for details of this sampling step.

Algorithm 1 below provides the pseudo-code of our proposed FBC algorithm, and we empirically confirm the convergence of this two-step MCMC algorithm in Section E.4. See Figure 6 in Section B.1 for a visualization of the algorithm.

---

**Algorithm 1** FBC (Fair Bayesian model-based Clustering) algorithm

---

1: **Inputs:** Data $(\mathcal{D} = \mathcal{D}^{(0)} \cup \mathcal{D}^{(1)})$, Maximum number of iterations for inference ($\max_{\text{iter}}$).
2: Initialize $\Phi^{(0)}, \mathbf{T}^{(0)}, \mathbf{T}_0^{(0)}, E^{(0)}$
3: **for** $t = 1$ to $\max_{\text{iter}}$ **do**
4:    (STEP 1) Propose $(\mathbf{T}', \mathbf{T}_0', E')$ and compute the acceptance probability $\alpha(\mathbf{T}', \mathbf{T}_0', E')$
5:    Sample $u \sim \text{Uniform}(0, 1)$
6:    **if** $u < \alpha(\mathbf{T}', \mathbf{T}_0', E')$ **then**
7:       Accept) $\mathbf{T}^{(t)} = \mathbf{T}', \mathbf{T}_0^{(t)} = \mathbf{T}_0', E^{(t)} = E'$
8:    **else**
9:       Reject) $\mathbf{T}^{(t)} = \mathbf{T}^{(t-1)}, \mathbf{T}_0^{(t)} = \mathbf{T}_0^{(t-1)}, E^{(t)} = E^{(t-1)}$
10:   **end if**
11:   (STEP 2) Sample $\Phi^{(t)}$ from posterior $p(\Phi|\mathbf{T}^{(t)}, \mathbf{T}_0^{(t)}, E^{(t)}, \mathcal{D})$
12: **end for**
13: **Return:** Posterior samples $\left\{\left(\Phi^{(t)}, \mathbf{T}^{(t)}, \mathbf{T}_0^{(t)}, E^{(t)}\right)\right\}_{t=1}^{\max_{\text{iter}}}$.

---

**Intuitive understanding of FBC** From an optimization perspective, our Bayesian formulation can be viewed as an analogue of a cost-fairness decomposition: (i) the likelihood plays the role of the clustering cost, and (ii) the fair prior space plays a role of the fairness constraint. In other words, we design the fair-constrained space via the prior, while our MCMC algorithm stochastically searches for both the parameters and the matchings within this fair-constrained space.

## 5 EXPERIMENTS

In this section, we empirically show that FBC (i) performs competitive to existing baselines in terms of the trade-off between clustering utility and fairness level; (ii) infers the number of clusters $K$ reasonably well; (iii) is easily applicable when data contain both continuous and categorical variables; (iv) can perform well under certain constraints on the cluster sizes.

**Datasets and performance measures** We analyze three real benchmark datasets: ADULT (Becker & Kohavi, 1996), BANK (Moro et al., 2014), and DIABETES (Smith et al., 1988). All features in the datasets are continuous and so we use the Gaussian mixture model. We standardize all features of data to have zero mean and unit variance. Note that the three datasets include binary class labels.

For clustering utility, we consider the Cost (i.e., the average distance to the center of the assigned cluster from each data point), which is defined as: $\texttt{Cost} := \sum_{k=1}^{K} \sum_{X_i \in C_k} \|X_i - \hat{\mu}_k\|^2 / n$, where $C_k := C_k^{(0)} \cup C_k^{(1)}$ is the set of data assigned to the $k^{\text{th}}$ cluster. Here, $\hat{\mu}_k := \sum_{X_i \in C_k} X_i / |C_k|$ denotes the center of the $k^{\text{th}}$ cluster. We additionally consider a density-based measure NLD (Negative Log Density). For baseline methods, we fit Gaussian for each cluster, then compute the weighted sum of negative log-likelihood. For FBC, we report the negative log posterior density for NLD. As the datasets include binary class labels, we also evaluate Acc (classification accuracy). To calculate Acc, we assign labels as follows: for each cluster, we set the label of every element to the cluster's majority label. Lower Cost / Lower NLD / Higher Acc imply a better clustering utility.

For fairness level, we consider two measures: (i) $\Delta(\boldsymbol{Z})$ defined in Equation (6) and (ii) the balance (Bal) defined as $\texttt{Bal} := \min_{k \in [K]} \texttt{Bal}_k$, where $\texttt{Bal}_k := \min\{|C_k^{(0)}|/|C_k^{(1)}|, |C_k^{(1)}|/|C_k^{(0)}|\}$ which is popularly considered in recent fair clustering literature (Backurs et al., 2019; Ziko et al., 2021; Esmaeili et al., 2021; Kim et al., 2025b). We abbreviate $\Delta(\boldsymbol{Z})$ by $\Delta$ in this section. See Section E.1 for details about the datasets and measures.

**Baselines and implementation details** For a fairness-agnostic method, we consider the Mixtures of Finite Mixtures (MFM) algorithm proposed by (Miller & Harrison, 2018). For baseline fair clustering methods, we consider several existing non-Bayesian approaches: SFC (Backurs et al., 2019), VFC (Ziko et al., 2021), and FCA (Kim et al., 2025b). SFC is a scalable fairlet-based fair clustering method, VFC is an in-processing approach by adding a fairness regularizer, and FCA is a recent work using the matching map for fair $K$-means clustering. Whenever $r > 0$, we set $R$ (in Section 2.1.2) as a random subset of $[n_0]$ of size $r$. Omitted experimental details are given in Section E.2.

### 5.1 FAIR CLUSTERING PERFORMANCE (KNOWN $K$)

**Comparison of FBC to existing fair clustering algorithms** We investigate whether FBC yields reasonable clustering results compared to baselines when $K$ is fixed at $k_*$, by assessing the trade-off between utility (Cost, NLD, and Acc) and fairness ($\Delta$ and Bal). That is, we set $p_K(k) = \mathbb{I}(K = k_*)$ in FBC. We run each algorithm to achieve the maximum fairness (e.g., $\Delta \approx 0$) for a fair comparison. Table 1 shows the results, suggesting that FBC is competitive to the baselines when $K$ is fixed at $k_* = 10$. Detailed comparisons are given as follows.

First, compared to SFC and VFC: FBC yields superior utility than SFC (i.e., lower Cost and NLD and higher Acc) in almost all cases, while both methods achieve near-perfect fairness; FBC attains a higher level of fairness than VFC in every case (lower $\Delta$ and higher Bal), with only slight losses in utility; indeed, on ADULT dataset, FBC even achieves higher accuracy.

Second, compared to FCA: FBC performs highly competitive utility and fairness, whereas FCA and FBC are conceptually similar in the sense that both are based on matching map. Although FCA

directly minimizes cost whereas FBC focuses on the posterior density, it is surprising that FBC attains competitive `Costs`. This result suggests that the matching map $\mathbf{T}$ in our MCMC inference may be also optimal in frequentist-view of the `Cost` minimizing approach. Furthermore, Table 2 in Section E.3.1 shows that FBC requires less computation time (near 50% of FCA). This is because FCA optimizes $\mathbf{T}$ by solving linear program at each iteration, while FBC searches $\mathbf{T}$ stochastically.

Table 1: Comparison of utility (`Cost`, `NLD`, `Acc`) and fairness ($\Delta$, `Bal`) for $k_* = 10$. See Table 3 in Section E.3.1 for the similar results for $k_* = 2$. Bold-faced values and underlined values (`Cost` and $\Delta$) indicate the best and second-best values, respectively, among the methods that achieve near-perfect fairness levels.

| Dataset | Measure | | SFC | VFC | FCA | FBC ✓ |
|---|---|---|---|---|---|---|
| ADULT | UTILITY | Cost (↓) | 3.129 | 1.702 | **1.869** | 1.954 |
| | | NLD (↓) | 6.498 | 6.134 | 6.174 | 6.242 |
| | | Acc (↑) | 0.763 | 0.765 | 0.785 | 0.781 |
| | FAIRNESS | $\Delta$ (↓) | 0.006 | 0.039 | **0.001** | **0.001** |
| | | Bal (↑) | 0.491 | 0.277 | 0.493 | 0.491 |
| BANK | UTILITY | Cost (↓) | 2.868 | 1.552 | **1.785** | 1.787 |
| | | NLD (↓) | 6.828 | 6.270 | 6.357 | 6.302 |
| | | Acc (↑) | 0.893 | 0.893 | 0.891 | 0.890 |
| | FAIRNESS | $\Delta$ (↓) | 0.003 | 0.040 | **0.001** | 0.004 |
| | | Bal (↑) | 0.647 | 0.513 | 0.648 | 0.615 |
| DIABETES | UTILITY | Cost (↓) | 5.115 | 3.077 | **3.441** | 3.757 |
| | | NLD (↓) | 9.381 | 8.946 | 9.033 | 9.088 |
| | | Acc (↑) | 0.656 | 0.724 | 0.690 | 0.693 |
| | FAIRNESS | $\Delta$ (↓) | 0.057 | 0.217 | **0.004** | 0.013 |
| | | Bal (↑) | 0.824 | 0.083 | 0.929 | 0.882 |

**Fairness level control of FBC**   We also numerically confirm that the fairness level can be controlled by $m$ (the size of $E$). Figure 8 in Section E presents the trade-off between $m$ and the fairness levels $\Delta$ and `Bal`, showing that a smaller $m$ usually results in a fairer clustering.

## 5.2 REASONABLE INFERENCE OF $K$ (UNKNOWN $K$)

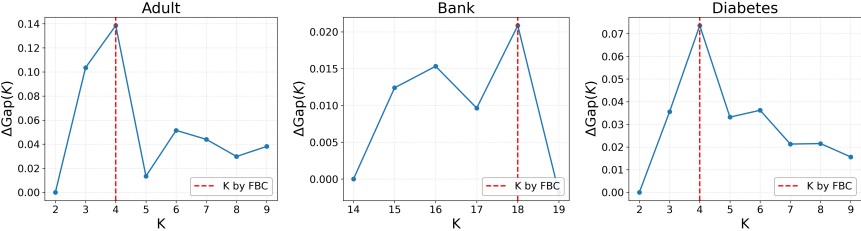

Figure 3: $K$ vs. $\Delta\mathrm{Gap}(K)$ of FCA on (left) ADULT, (center) BANK, and (right) DIABETES datasets. Red vertical dashed lines indicate the $K$ inferred by FBC.

**Cluster quality**   When $K$ is unknown, FBC treats it as a random variable and infer as $K = 4, 18$, and 4 for ADULT, BANK, and DIABETES datasets, respectively, and these are at the posterior modes (Figure 9 in Section E.3.2). To assess the optimality of the inferred $K$ in view of cluster quality, we consider the Gap statistic (Tibshirani et al., 2002), a well-known measure developed to determine the optimal $K$ based on within-cluster dispersions. We run FCA for various $K$s, calculate the Gap statistic for each $K$, and then plot $\Delta\mathrm{Gap}(K) = \mathrm{Gap}(K) - \mathrm{Gap}(K-1)$ for $K \geq 2$ with $\mathrm{Gap}(2) = 0$. The elbow, i.e., the largest increase, occurs at the $K$ inferred by FBC (see Figure 3).

We further evaluate the trade-off performance compared to the baseline methods. That is, we run baseline algorithms with $K$s inferred by FBC, then compare the trade-off performance. Table 5 in Section E.3.2 presents the results, suggesting that FBC is still competitive to the baselines.

**Density estimation** As FBC is a model-based approach, it can be used for not only clustering but also density estimation. Similar to the Gap statistic, Figure 4 shows the difference $\Delta\text{NLD}(K) = \text{NLD}(K) - \text{NLD}(K-1)$ in posterior densities (on test data) for various $K$s in FBC, which suggests that the inferred ones are optimal in view of density estimation (i.e., achieves the elbow in NLD on the test data). More details are given in E.3.2.

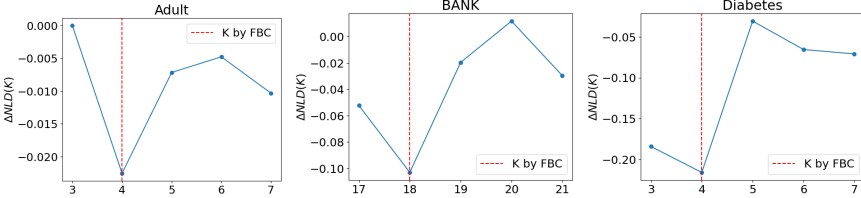

Figure 4: $K$ vs. $\Delta\text{NLD}(K)$ of FBC with fixed $K$ on (left) ADULT, (center) BANK, and (right) DIABETES datasets. Red vertical dashed lines indicate the $K$ inferred by FBC.

### 5.3 FLEXIBILITIES OF FBC

**Applicability to various data types** FBC accommodates both continuous and categorical data as long as the likelihood of the mixture model is defined. We empirically compare FBC with FairDen (Krieger et al., 2025), which likewise supports categorical data as well. Section E.3.3 details the analysis, where Table 6 shows that (i) using both continuous and categorical data improves the utility (Acc), highlighting the limitation of distance-based clustering methods that operate only on continuous data; and (ii) FBC outperforms FairDen in terms of the fairness–utility trade-off.

**Clustering under size constraints** FBC can also flexibly work under cluster size constraints (i.e., the upper bound of cluster size) with a slight modification to the inference algorithm. Section E.3.4 details how we incorporate the size constraint in FBC; results in Table 7 shows that under the size constraint, FBC achieves lower Cost while attaining perfect fairness, outperforming the post-processing method of Bera et al. (2019).

### 5.4 ABLATION STUDIES

We conduct ablation studies on three topics: (i) Impact of the temperature constant $\tau$ in the prior of $\mathbf{T}$ defined in Definition 3.1; (ii) Validity of the two proposed heuristic approaches of selecting $R$ in Section 2.1.2; (iii) Influence of the covariance structure in the Gaussian mixture model. Details and results are given in Section E.3.5, showing the robustness on $\tau$ and $R$, and that using a more complex covariance improves density estimation (lower NLD) while maintaining high fairness level.

## 6 CONCLUDING REMARKS

In this paper, we proposed a fair Bayesian mixture model for fair clustering with unknown number of clusters and developed an efficient MCMC algorithm called FBC. A key advantage of FBC is its ability to infer the number of clusters.

A problem not pursued in this work is the assignment of new data to the learned clusters. FBC requires matched instances to be assigned to the same cluster, but the inferred matching map is only available for the training data. To address this, one may approximate the learned matching map using a parametric model and apply it to new test data. We leave this approach as future work.

**Ethics Statement**    The fairness notion we consider in this study, i.e., group (or proportional) fairness, is widely examined in recent literature. All the datasets we consider are publicly available. We believe this research can contribute to avoid discriminatory results in clustering, rather than creating new ethical concerns.

**Reproducibility Statement**    For theoretical results, we document full proofs and mathematical definitions for the stated theorems, in Appendix. For experimental results, we state implementation details across the main body and Appendix, and further include the source files in the supplementary materials.

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

# APPENDIX

## A  DETAILS AND THEORIES FOR SECTION 2

### A.1  HANDLING THE CASE OF $r > 0$

As explained in Section 2.1.2, we need to resolve the issue that the matching map $\mathbf{T}$ does not always corresponds to a fair $\mathbf{Z} \in \mathcal{Z}_0^{\text{Fair}}$, for the case of $r > 0$. That is, there exists $\mathbf{Z}$ that satisfies $Z_j^{(1)} = Z_{\mathbf{T}(j)}^{(0)}$ for some $\mathbf{T} \in \mathcal{T}$ but is not fair. In turn, a sufficient condition of $\mathbf{Z}$ with $Z_j^{(1)} = Z_{\mathbf{T}(j)}^{(0)}$ for some $\mathbf{T} \in \mathcal{T}$ to be fair is that $|C_k^{(0)}|/n_0 = |C_k^{(0)} \cap R_{\mathbf{T}}|/r$, where $C_k^{(0)} = \{i : Z_i^{(0)} = k\}$ for $k \in [K]$ (see Proposition A.2 for the proof). That is, fairness of $\mathbf{Z}$ depends on both $\mathbf{T}$ and $\mathbf{Z}^{(0)}$, which would make the computation of the posterior inference expensive.

Let $\mathbb{P}_{n_0}$ and $\mathbb{P}_R$ be the empirical distributions of $\{X_i^{(0)}, i \in [n_0]\}$ and $\{X_i^{(0)}, i \in R\}$, respectively. If $\mathbb{P}_{n_0}(\cdot) = \mathbb{P}_R(\cdot)$, we have $|C_k^{(0)}|/n_0 = |C_k^{(0)} \cap R|/r$, and thus any fair $\mathbf{Z}$ belongs to $\mathcal{Z}_0^{\text{Fair}}$. This observation suggests us to choose $R$ such that $\{X_i^{(0)}, i \in R\}$ represents the original data $\{X_i^{(0)}, i \in [n_0]\}$ well. Hence, as previously stated, we suggest two candidates of $R$, (i) a random subset and (ii) cluster centers. For (ii), the cluster centers can be found by $K$-medoids algorithm.

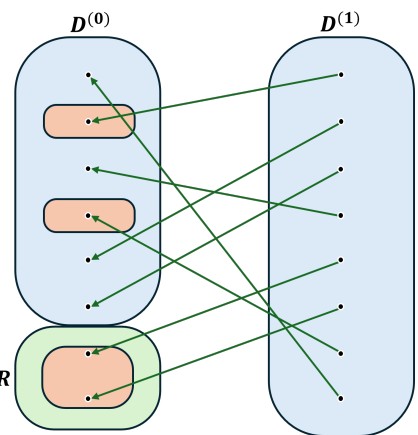

Figure 5: An illustration on the modification for the case of $r > 0$. Orange-indicated points are upsampled indices from $[n_0]$ to construct $R$.

### A.2  PROOFS

**Proposition 2.1** Assume that $n_0 = n_1 = \bar{n}$. Then, we have: $\mathbf{Z} \in \mathcal{Z}_0^{\text{Fair}} \iff$ There exists a matching map $\mathbf{T}$ such that $Z_j^{(1)} = Z_{\mathbf{T}(j)}^{(0)}, \forall j \in [\bar{n}]$.

*Proof of Proposition 2.1.* See the proof of Proposition 2.2, as this proposition is a special case of Proposition 2.2, with $\beta = 1$. $\qquad\qquad\square$

**Proposition 2.2** Assume that $n_1 = \beta n_0$ for some positive integer $\beta$. Then, we have: $\mathbf{Z} \in \mathcal{Z}_0^{\text{Fair}} \iff$ There exists a matching map $\mathbf{T}$ such that $Z_j^{(1)} = Z_{\mathbf{T}(j)}^{(0)}, \forall j \in [n_1]$.

*Proof of Proposition 2.2.* ( $\implies$ ) Recall that $C_k^{(0)} = \{i : Z_i^{(0)} = k\}$ and $C_k^{(1)} = \{j : Z_j^{(1)} = k\}$. Note that $\mathbf{Z} \in \mathcal{Z}_0^{\text{Fair}}$ implies $\beta|C_k^{(0)}| = |C_k^{(1)}|$ for all $k \in [K]$. Hence, for all $k \in [K]$, there exists an onto map $\mathbf{T}_k$ from $C_k^{(1)}$ to $C_k^{(0)}$ such that $|\mathbf{T}_k^{-1}(i)| = \beta$ for all $i \in C_k^{(0)}$. Letting $\mathbf{T}(j) := \sum_{k=1}^{K} \mathbf{T}_k(j)\mathbb{I}(j \in C_k^{(1)}), j \in [n_1]$ concludes the proof.

( $\Longleftarrow$ ) Suppose that there exists a matching map $\mathbf{T}$ such that $Z_j^{(1)} = Z_{\mathbf{T}(j)}^{(0)}$ for all $j \in [n_1]$. Then, we have $\beta |C_k^{(0)}| = |C_k^{(1)}|$ for all $k \in [K]$. Hence, $\sum_{i=1}^{n_0} \mathbb{I}(Z_i^{(0)} = k)/n_0 = |C_k^{(0)}|/n_0 = \beta |C_k^{(0)}|/n_1 = |C_k^{(1)}|/n_1 = \sum_{j=1}^{n_1} \mathbb{I}(Z_j^{(1)} = k)/n_1$, which implies $\mathbf{Z} \in \mathcal{Z}_0^{\text{Fair}}$. $\qquad\square$

Propositions A.1 and A.2 below support the claims in Section 2.1.2. For enhanced readability, we first recall some notations/assumptions:

- $n_1 = \beta n_0 + r$.
- $\mathbf{T} : [n_1] \to [n_0]$ is a function such that it is onto, $|\mathbf{T}^{-1}(i)|$ is either $\beta$ or $\beta+1$ and $|R_{\mathbf{T}}| = r$, where $R_{\mathbf{T}} = \{i : |\mathbf{T}^{-1}(i)| = \beta+1\}$. Let $\mathcal{T}$ be the set of all such matching maps (functions satisfying these conditions).
- Let $R_{\mathbf{T}} = \{i : |\mathbf{T}^{-1}(i)| = \beta + 1\}$ for a given $\mathbf{T} : [n_1] \to [n_0]$.
- For given $Z_i^{(0)}, i \in [n_0]$ and $Z_j^{(1)}, j \in [n_1]$, we define $C_k^{(0)} = \{i : Z_i^{(0)} = k\}$ and $C_k^{(1)} = \{j : Z_j^{(1)} = k\}$ for $k \in [K]$.

**Proposition A.1.** *Assume that $r > 0$. For a given fair $\mathbf{Z} \in \mathcal{Z}_0^{\text{Fair}}$, there exists a function $\mathbf{T} \in \mathcal{T}$.*

*Proof of Proposition A.1.* Since $\mathbf{Z} \in \mathcal{Z}^{\text{Fair}}$, for all $k \in [K]$ we have $|C_k^{(1)}| = \frac{n_1}{n_0} |C_k^{(0)}|$, which is an integer. Let $k^* = \arg\min_k |C_k^{(0)}|$ and $\alpha = \frac{r |C_{k^*}^{(0)}|}{n_0} \in \mathbb{N}$. Then, we can construct $\mathbf{T}_{k^*} : C_{k^*}^{(1)} \to C_{k^*}^{(0)}$ so that exactly $\alpha$ elements of $C_{k^*}^{(0)}$ have preimage size $\beta + 1$, and the remaining $|C_{k^*}^{(0)}| - \alpha$ have size $\beta$.

Next, for each $l \neq k^*$, set $a_l = \frac{|C_l^{(0)}|}{|C_{k^*}^{(0)}|} \in \mathbb{N}$, so that $|C_l^{(0)}| = a_l |C_{k^*}^{(0)}|$ and $|C_l^{(1)}| = a_l |C_{k^*}^{(1)}|$. Define $\alpha_l = a_l \alpha$, and similarly construct $\mathbf{T}_l : C_l^{(1)} \to C_l^{(0)}$ so that exactly $\alpha_l$ points in $C_l^{(0)}$ have preimage size $\beta + 1$, and the remaining $|C_l^{(0)}| - \alpha_l$ have size $\beta$.

Finally, let $\mathbf{T}(j) := \sum_{k=1}^K \mathbf{T}_k(j) \mathbb{I}(j \in C_k^{(1)}), j \in [n_1]$. Then, $\mathbf{T}$ is onto, each $\mathbf{T}_k$ has size $\beta$ or $\beta + 1$, and $|R_{\mathbf{T}}| = \sum_{k=1}^K \alpha_k = \alpha \sum_{k=1}^K a_k = \alpha \cdot \frac{n_0}{|C_{k^*}^{(0)}|} = \frac{r |C_{k^*}^{(0)}|}{n_0} \cdot \frac{n_0}{|C_{k^*}^{(0)}|} = r$, which completes the proof. $\qquad\square$

**Proposition A.2.** *Assume that $r > 0$. For a given $\mathbf{T}$, let $\mathbf{Z}$ with $Z_j^{(1)} = Z_{\mathbf{T}(j)}^{(0)}, j \in [n_1]$. If $\mathbf{T} \in \mathcal{T}$ satisfied $|C_k^{(0)}|/n_0 = |C_k^{(0)} \cap R_{\mathbf{T}}|/r$ for all $k \in [K]$, we have that $\mathbf{Z} \in \mathcal{Z}_0^{\text{Fair}}$.*

*Proof of Proposition A.2.* Fix a $k \in [K]$. By definition of $\mathbf{Z}$ that $C_k^{(1)} = \{j : Z_j^{(1)} = k\} = \{j : Z_{\mathbf{T}(j)}^{(0)} = k\}$, we have that

$$|C_k^{(1)}| = \sum_{i \in C_k^{(0)}} |\mathbf{T}^{-1}(i)| = \sum_{i \in C_k^{(0)}} (\beta + \mathbb{I}(i \in R_{\mathbf{T}})) = \beta |C_k^{(0)}| + |C_k^{(0)} \cap R_{\mathbf{T}}|.$$

By the sufficient condition $|C_k^{(0)} \cap R_{\mathbf{T}}| = (r/n_0) |C_k^{(0)}|$ and since $n_1 = \beta n_0 + r$, it follows that

$$|C_k^{(1)}| = \beta |C_k^{(0)}| + \frac{r}{n_0} |C_k^{(0)}| = \frac{\beta n_0 + r}{n_0} |C_k^{(0)}| = \frac{n_1}{n_0} |C_k^{(0)}|,$$

which concludes the proof. $\qquad\square$

**Proposition A.3.** *Assume that $r > 0$. Let $\mathbf{T} \in \mathcal{T}$ and define $\mathbf{Z}$ by $Z_j^{(1)} = Z_{\mathbf{T}(j)}^{(0)}, j \in [n_1]$. Then, the fairness level $\Delta(\mathbf{Z})$ is uniformly bounded as*

$$\Delta(\mathbf{Z}) \leq \frac{r(n_0 - r)}{n_0 n_1} = \max_{\mathbf{Z}} \Delta(\mathbf{Z}),$$

*where the maximum (i.e., the worst-case) is attained when all instances in $R_{\mathbf{T}}$ are assigned to the same cluster.*

*Proof.* For $k \in [K]$, let $C_k^{(1)} = \{j : Z_j^{(1)} = k\}$ and define $p_k^{(0)} := \frac{|C_k^{(0)}|}{n_0}$ and $p_k^{(1)} := \frac{|C_k^{(1)}|}{n_1}$. By the definition of $\boldsymbol{Z}$, we have $C_k^{(1)} = \{j : Z_j^{(1)} = k\} = \{j : Z_{\mathbf{T}(j)}^{(0)} = k\}$, so

$$|C_k^{(1)}| = \sum_{i \in C_k^{(0)}} \left|\mathbf{T}^{-1}(i)\right| = \sum_{i \in C_k^{(0)}} \left(\beta + \mathbb{I}(i \in R_{\mathbf{T}})\right) = \beta \left|C_k^{(0)}\right| + |C_k^{(0)} \cap R_{\mathbf{T}}|.$$

Hence

$$p_k^{(0)} - p_k^{(1)} = \frac{|C_k^{(0)}|}{n_0} - \frac{\beta |C_k^{(0)}| + |C_k^{(0)} \cap R_{\mathbf{T}}|}{n_1} = \frac{r}{n_1} p_k^{(0)} - \frac{|C_k^{(0)} \cap R_{\mathbf{T}}|}{n_1}. \tag{18}$$

Define $q_k := \frac{|C_k^{(0)} \cap R_{\mathbf{T}}|}{r}, k \in [K]$. Then $q_k \geq 0$ and $\sum_{k=1}^{K} q_k = 1$, so $\mathbf{q} = (q_1, \ldots, q_K)$ is a probability vector. Then, from Equation (18) we can write

$$p_k^{(0)} - p_k^{(1)} = \frac{r}{n_1}\left(p_k^{(0)} - q_k\right), k \in [K], \tag{19}$$

and therefore we have

$$\Delta(\boldsymbol{Z}) = \frac{1}{2}\sum_{k=1}^{K}\left|p_k^{(0)} - p_k^{(1)}\right| = \frac{r}{2n_1}\sum_{k=1}^{K}\left|p_k^{(0)} - q_k\right| = \frac{r}{2n_1}\|p^{(0)} - q\|_1, \tag{20}$$

where $\mathbf{p}^{(0)} = (p_1^{(0)}, \ldots, p_K^{(0)})^\top$. Since $\mathbf{p}^{(0)}$ and $\mathbf{q}$ are probability vectors, we have $\|\mathbf{p}^{(0)} - \mathbf{q}\|_1 \leq 2$. and thus $\Delta(\boldsymbol{Z}) \leq \frac{r}{n_1}$.

Moreover, the $\ell_1$ distance $\|\mathbf{p}^{(0)} - \mathbf{q}\|_1$ is maximized over all probability vectors $\mathbf{q}$ at an extreme point of the simplex, that is, at some $\mathbf{q} = e_{k^*}$ with $(\mathbf{e}_{k^*})_{k^*} = 1$ and $(\mathbf{e}_{k^*})_k = 0$ for $k \neq k^*$. Such a vector corresponds exactly to the case where all indices in $R_{\mathbf{T}}$ belong to a single cluster $C_{k^*}^{(0)}$, that is, $R_{\mathbf{T}} \subset C_{k^*}^{(0)}$. Hence this configuration yields the worst-case (maximal) value of $\Delta(\boldsymbol{Z})$, so it suffices to focus on this case.

Under $R_{\mathbf{T}} \subset C_{k^*}^{(0)}$, we have $|C_{k^*}^{(0)} \cap R_{\mathbf{T}}| = r, |C_k^{(0)} \cap R_{\mathbf{T}}| = 0$ for $k \neq k^*$. Plugging this into Equation (18) yields $p_{k^*}^{(0)} - p_{k^*}^{(1)} = \frac{r}{n_1}p_{k^*}^{(0)} - \frac{r}{n_1} = \frac{r}{n_1}\left(p_{k^*}^{(0)} - 1\right)$, and for $k \neq k^*$, we have $p_k^{(0)} - p_k^{(1)} = \frac{r}{n_1}p_k^{(0)}$. Therefore, we get $\left|p_{k^*}^{(0)} - p_{k^*}^{(1)}\right| = \frac{r}{n_1}\left|p_{k^*}^{(0)} - 1\right| = \frac{r}{n_1}\left(1 - p_{k^*}^{(0)}\right)$, and for $k \neq k^*$, we have $\left|p_k^{(0)} - p_k^{(1)}\right| = \frac{r}{n_1}p_k^{(0)}$. Using the definition of $\Delta(\boldsymbol{Z})$,

$$\Delta(\boldsymbol{Z}) = \frac{1}{2}\sum_{k=1}^{K}\left|p_k^{(0)} - p_k^{(1)}\right| = \frac{1}{2}\left(\frac{r}{n_1}\left(1 - p_{k^*}^{(0)}\right) + \sum_{k \neq k^*}\frac{r}{n_1}p_k^{(0)}\right).$$

Since $\sum_{k \neq k^*} p_k^{(0)} = 1 - p_{k^*}^{(0)}$, we obtain

$$\Delta(\boldsymbol{Z}) = \frac{1}{2} \cdot \frac{r}{n_1}\left(1 - p_{k^*}^{(0)} + 1 - p_{k^*}^{(0)}\right) = \frac{r}{n_1}\left(1 - p_{k^*}^{(0)}\right).$$

Finally, because $R_{\mathbf{T}} \subset C_{k^*}^{(0)}$ and $|R_{\mathbf{T}}| = r$, we have $|C_{k^*}^{(0)}| \geq r$, so that $p_{k^*}^{(0)} = \frac{|C_{k^*}^{(0)}|}{n_0} \geq \frac{r}{n_0}$. Hence

$$\Delta(\boldsymbol{Z}) = \frac{r}{n_1}\left(1 - p_{k^*}^{(0)}\right) \leq \frac{r}{n_1}\left(1 - \frac{r}{n_0}\right),$$

which completes the proof. $\qquad\square$

### A.3 FOR NON-PERFECT FAIRNESS

**Proposition A.4.** *Assume that $r = 0$. Let $\mathbf{T}$ and $\mathbf{T}_0$ be the maps defined in Section 2.2 for a given $m \leq n_1$. Then, for any $\boldsymbol{Z}$ satisfying $Z_j^{(1)} = Z_{\mathbf{T}(j)}^{(0)}, j \in [n_1] \setminus E$ and $Z_j^{(1)} = Z_{\mathbf{T}_0(j)}^{(0)}, j \in E$, we have that $\boldsymbol{Z} \in \mathcal{Z}_{m/n_1}^{\text{Fair}}$.*

*Proof of Proposition A.4.* We first note the following two facts: (i) $\sum_{k=1}^{K} |\frac{1}{n_0} \sum_{i \in [n_0]} \mathbb{I}(Z_i^{(0)} = k) - \frac{1}{n_1} \sum_{j \in [n_1]} \mathbb{I}(Z_{\mathbf{T}(j)}^{(0)} = k)| = 0$ for any given $\mathbf{T}$ due to the assumption $r = 0$. (ii) For any $\mathbf{T}$ and $\mathbf{T}_0$, there exist nonnegative integers $m_1, \ldots, m_K$ with $\sum_k m_k = 2m$ such that for each $k$, $|\sum_{j \in E} \mathbb{I}(Z_{\mathbf{T}(j)}^{(0)} = k) - \sum_{j \in E} \mathbb{I}(Z_{\mathbf{T}_0(j)}^{(0)} = k)| = m_k$. Therefore,

$$\Delta(\boldsymbol{Z}) = \frac{1}{2} \sum_{k=1}^{K} \left| \frac{1}{n_0} \sum_{i \in [n_0]} \mathbb{I}(Z_i^{(0)} = k) - \frac{1}{n_1} \sum_{j \in [n_1]} \mathbb{I}(Z_j^{(1)} = k) \right|$$

$$= \frac{1}{2} \sum_{k=1}^{K} \left| \frac{1}{n_0} \sum_{i \in [n_0]} \mathbb{I}(Z_i^{(0)} = k) - \frac{1}{n_1} \sum_{j \in [n_1] \setminus E} \mathbb{I}(Z_j^{(1)} = k) - \frac{1}{n_1} \sum_{j \in E} \mathbb{I}(Z_j^{(1)} = k) \right|$$

$$= \frac{1}{2} \sum_{k=1}^{K} \left| \frac{1}{n_0} \sum_{i \in [n_0]} \mathbb{I}(Z_i^{(0)} = k) - \frac{1}{n_1} \sum_{j \in [n_1] \setminus E} \mathbb{I}(Z_{\mathbf{T}(j)}^{(0)} = k) - \frac{1}{n_1} \sum_{j \in E} \mathbb{I}(Z_{\mathbf{T}_0(j)}^{(0)} = k) \right|$$

$$\leq \frac{1}{2} \sum_{k=1}^{K} \left| \frac{1}{n_0} \sum_{i \in [n_0]} \mathbb{I}(Z_i^{(0)} = k) - \frac{1}{n_1} \sum_{j \in [n_1]} \mathbb{I}(Z_{\mathbf{T}(j)}^{(0)} = k) \right|$$

$$+ \frac{1}{2} \sum_{k=1}^{K} \left| \frac{1}{n_1} \sum_{j \in E} \mathbb{I}(Z_{\mathbf{T}(j)}^{(0)} = k) - \frac{1}{n_1} \sum_{j \in E} \mathbb{I}(Z_{\mathbf{T}_0(j)}^{(0)} = k) \right|$$

$$= 0 + \frac{1}{2} \frac{1}{n_1} \sum_{k=1}^{K} m_k = \frac{m}{n_1}.$$

Thus $\Delta(\boldsymbol{Z}) \leq m/n_1$, so $\boldsymbol{Z} \in \mathcal{Z}_{m/n_1}^{\text{Fair}}$. $\qquad\square$

### A.4 RELATIONSHIP BETWEEN $\Delta$ AND BAL

Two fairness measures - $\Delta$ and $\mathtt{Bal}$ - that we consider in this work are closely related, as proven in Kim et al. (2025b). For readers' sake, we provide a rigorous statement below.

**Proposition A.5** (Proposition 4.2. of Kim et al. (2025b))**.** *Suppose $n_0 \leq n_1$. For given cluster assignments $Z_i^{(0)}, i \in [n_0]$ and $Z_j^{(1)}, j \in [n_1]$, we have $\max_{k \in [K]} |\frac{\sum_{i=1}^{n_0} \mathbb{I}(Z_i^{(0)} = k)/n_0}{\sum_{j=1}^{n_1} \mathbb{I}(Z_j^{(1)} = k)/n_1} - \frac{n_0}{n_1}| \leq c\Delta$, where $c = \frac{n_0}{n_1} \max_{k \in [K]} \frac{n_1}{\sum_{j=1}^{n_1} \mathbb{I}(Z_j^{(1)} = k)}$. This implies that the difference between balance and the balance of perfect fairness ($= n_0/n_1$) is bounded by $\Delta$.*

Furthermore, experimental results in Section 5 also numerically support the use of $\Delta$ in the sense that controlling $\Delta$ effectively controls $\mathtt{Bal}$.

# B DETAILS OF FBC

## B.1 HIGH-LEVEL UNDERSTANDING

**Flow diagram** We provide a flow diagram of FBC in Figure 6, which is a visualization of Algorithm 1. Note that it focuses on the perfect fairness case with $n_0 = n_1$ for simplicity. FBC iteratively obtains MCMC samples by a Gibbs sampler: STEP 1 provides a matching function at the time step $(t)$, and STEP 2 yields a mixture parameter at the time step $(t)$.

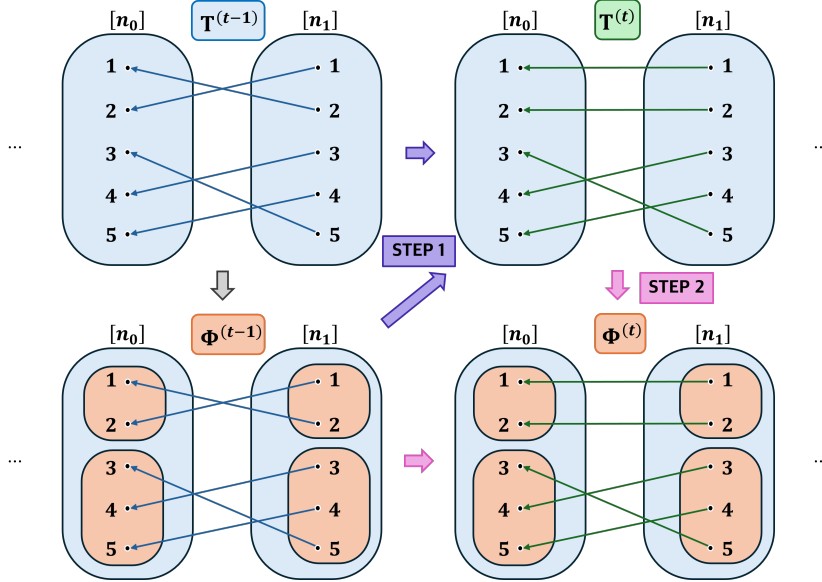

Figure 6: A flow diagram of FBC.

**Understanding FBC in view of fairlets** For simplicity, let ignore $\mathbf{T}_0$ and $E$, and consider only the perfect-fairness case with $\mathbf{T}$. In this setting, $\mathbf{T}$ can be seen as a map to build fairlets, and we treat $\mathbf{T}$ as a random variable to perform posterior inference. Inspired by MFM (Miller & Harrison, 2018) which obtains MCMC samples of model parameters, we integrate the MH algorithm into the original MCMC procedure to yield posterior samples of $\mathbf{T}$. We also theoretically validate this proposed algorithm in Section B.3. Moreover, for reasonable initialization and prior for $\mathbf{T}$, we incorporate ideas from optimal transport inspired of the the fairlet-based approaches. A crucial difference is that, we update $\mathbf{T}$ for higher clustering utility (e.g., higher log-likelihood or lower clustering cost) rather than using a fixed $\mathbf{T}$, as fairlet-based approaches do.

## B.2 DETAILS OF STEP 1 (SAMPLING $\mathbf{T}, \mathbf{T}_0, E$) IN SECTION 4

In this section, we first provide a detailed explanation about the calculation of $\alpha'(\mathbf{T}', \mathbf{T}'_0, E)$ in Equation (17). First, we have that

$$\alpha'(\mathbf{T}', \mathbf{T}'_0, E') = \frac{p(\mathbf{T}', \mathbf{T}'_0, E'|\Phi, \mathcal{D})q((\mathbf{T}', \mathbf{T}'_0, E') \to (\mathbf{T}, \mathbf{T}_0, E))}{p(\mathbf{T}, \mathbf{T}_0, E|\Phi, \mathcal{D})q((\mathbf{T}, \mathbf{T}_0, E) \to (\mathbf{T}', \mathbf{T}'_0, E'))}$$

$$= \frac{p(\mathbf{T}', \mathbf{T}'_0, E')\mathcal{L}(\mathcal{D}; \Phi, \mathbf{T}', \mathbf{T}'_0, E')q((\mathbf{T}', \mathbf{T}'_0, E') \to (\mathbf{T}, \mathbf{T}_0, E))}{p(\mathbf{T}, \mathbf{T}_0, E)\mathcal{L}(\mathcal{D}; \Phi, \mathbf{T}, \mathbf{T}_0, E)q((\mathbf{T}, \mathbf{T}_0, E)) \to (\mathbf{T}', \mathbf{T}'_0, E'))}$$

$$= \frac{e(\mathbf{T}')\mathcal{L}(\mathcal{D}; \Phi, \mathbf{T}', \mathbf{T}'_0, E')}{e(\mathbf{T})\mathcal{L}(\mathcal{D}; \Phi, \mathbf{T}, \mathbf{T}_0, E)},$$

where the last equality holds since

$$q((\mathbf{T}', \mathbf{T}'_0, E') \to (\mathbf{T}, \mathbf{T}_0, E)) = q((\mathbf{T}, \mathbf{T}_0, E) \to (\mathbf{T}', \mathbf{T}'_0, E'))$$

and $p(\mathbf{T}', \mathbf{T}'_0, E')/p(\mathbf{T}, \mathbf{T}_0, E) = \mathbf{e}(\mathbf{T}')/\mathbf{e}(\mathbf{T})$, because the priors of $\mathbf{T}_0$ and $E$ are uniform.

For the likelihood ratio, if we use a conjugate prior which enables the calculation of marginal likelihood, we have

$$\frac{\mathcal{L}(\mathcal{D}; \Phi, \mathbf{T}', \mathbf{T}_0', E')}{\mathcal{L}(\mathcal{D}; \Phi, \mathbf{T}, \mathbf{T}_0, E)} = \frac{\prod_{c \in \mathcal{C}} m(X^c | \mathbf{T}', \mathbf{T}_0', E')}{\prod_{c \in \mathcal{C}} m(X^c | \mathbf{T}, \mathbf{T}_0, E)},$$

where

$$m(X^c | \mathbf{T}, \mathbf{T}_0, E) = \int_\Theta \left[ \prod_{i \in c} f(X_i^{(0)} | \phi_c) \prod_{j \in J(c; \mathbf{T}, \mathbf{T}_0, E)} f(X_j^{(1)} | \phi_c) \right] H(d\theta), \qquad (21)$$

$J(c; \mathbf{T}, \mathbf{T}_0, E) := \{j \in E : \mathbf{T}_0(j) \in c\} \cup \{j \in [n_1] \setminus E : \mathbf{T}(j) \in c\}$ for $c \in \mathcal{C}$ and $X^c = \{X_i^{(0)} : i \in c\} \cup \{X_j^{(1)} : j \in J(c; \mathbf{T}, \mathbf{T}_0, E)\}$ for $c \in \mathcal{C}$. For a non-conjugate $H$, we have

$$\frac{\mathcal{L}(\mathcal{D}; \Phi, \mathbf{T}', \mathbf{T}_0', E')}{\mathcal{L}(\mathcal{D}; \Phi, \mathbf{T}, \mathbf{T}_0, E)} = \frac{\prod_{c \in \mathcal{C}} \left[ \prod_{i \in c} f(X_i^{(0)} | \phi_c) \prod_{j \in J(c; \mathbf{T}', \mathbf{T}_0', E')} f(X_j^{(1)} | \phi_c) \right]}{\prod_{c \in \mathcal{C}} \left[ \prod_{i \in c} f(X_i^{(0)} | \phi_c) \prod_{j \in J(c; \mathbf{T}, \mathbf{T}_0, E)} f(X_j^{(1)} | \phi_c) \right]}.$$

Note that these calculations are derived from the equivalent representation in Section 3.1.

Next, we provide an illustration of the proposal from STEP 1 in Figure 7. $\mathbf{T}'$ of the green line is randomly swapped from $\mathbf{T}$ of the blue line. $\mathbf{T}_0'$ of the orange line is a random proposal. $E'$ of the pink region is a random proposal of the given size. The final matching is visualized as the lines of the final diagram.

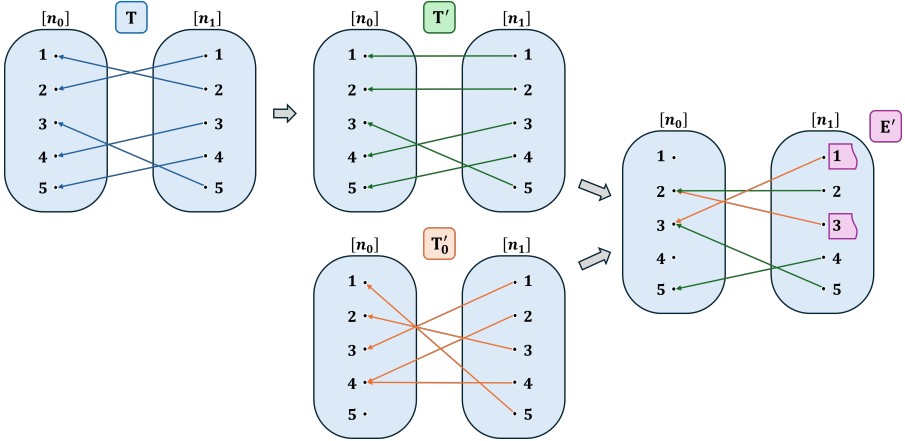

Figure 7: An illustration of the proposal from STEP 1.

## B.3 DETAILS OF STEP 2 (SAMPLING $\Phi$) IN SECTION 4

Here, we consider the following two cases. When $m(X^c | \mathbf{T}, \mathbf{T}_0, E)$ can be easily computed (e.g., $H$ is a conjugate prior), we sample $\mathcal{C} \sim p(\mathcal{C} | \mathcal{D}, \mathbf{T}, \mathbf{T}_0, E)$. Otherwise, when $m(X^c | \mathbf{T}, \mathbf{T}_0, E)$ is intractable, we sample $(\mathcal{C}, \phi) \sim p(\mathcal{C}, \phi | \mathcal{D}, \mathbf{T}, \mathbf{T}_0, E)$. If the marginal likelihood is computable, a direct adaptation of Algorithm 3 from Neal (2000); MacEachern & Müller (1998) is applicable. Otherwise, when the marginal likelihood is not computable, Algorithm 8 from Neal (2000) can be applied. Wherever the meaning is clear, we abbreviate $J(c; \mathbf{T}, \mathbf{T}_0, E)$ by $J(c)$ in this section.

**Conjugate prior** The modification of Algorithm 3 for FBC is described as below.

1. Initialize $\mathcal{C} = \{[n_0]\}$ (i.e., a single cluster)

2. Repeat the following steps $N$ times, to obtain $N$ samples. For $i = 1, \ldots, n_0$: Remove element $i \in [n_0]$ and its matched elements in $J(\{i\}; \mathbf{T}, \mathbf{T}_0, E) := \{j \in E : \mathbf{T}_0(j) = i\} \cup \{j \in [n_1] \setminus E : \mathbf{T}(j) = i\}$ from $\mathcal{C}$. Then, place them

- to $c' \in \mathcal{C} \setminus i$ with probability

$$\propto (|c'| + \gamma)\frac{m(X^{c'} \cup X^{\{i\}}|\mathbf{T}, \mathbf{T}_0, E)}{m(X^{c'}|\mathbf{T}, \mathbf{T}_0, E)}$$

where $X^{\{i\}} := \{X_i^{(0)}\} \cup \{X_j^{(1)} : j \in J(\{i\})\}$.

- to a new cluster with probability

$$\propto \gamma\frac{V_{n_0}(t+1)}{V_{n_0}(t)}m(X^{\{i\}}|\mathbf{T}, \mathbf{T}_0, E)$$

where $t$ is a number of clusters when $X^{\{i\}}$ are removed.

**Proposition B.1.** *The above modification of Algorithm 3 of Neal (2000) is a valid Gibbs sampler.*

*Proof of Proposition B.1.* The posterior density can be formulated as follows:

$$p\left(\mathcal{C}|\mathcal{D}, \mathbf{T}, \mathbf{T}_0, E\right) \propto p(\mathcal{C}) \cdot p\left(X_{1:n_0}^{(0)}, X_{1:n_1}^{(1)}|\mathcal{C}, \mathbf{T}, \mathbf{T}_0, E\right)$$
$$= V_{n_0}(t)\prod_{c \in \mathcal{C}}\gamma^{(|c|)} \cdot \prod_{c \in \mathcal{C}}m(X^c|\mathbf{T}, \mathbf{T}_0, E)$$

To justify the proposed modification of Algorithm 3, we only need to calculate the probabilities of placing $X_i^{(0)}$ and its matched elements $\{X_j^{(1)}; j \in J(c(i))\}$: (i) to an existing partition $c'$, or (ii) to a new cluster. Let $\mathcal{C}_{-i}$ be the collection of clusters where $X_i^{(0)}$ and its matched elements $\{X_j^{(1)} : j \in J(\{i\})\}$ are removed from $\mathcal{C}$. The calculation can be done as follows:

(i) to existing $c'$:

The term $\gamma^{(|c'|)}$ in the prior term $p(\mathcal{C}) \propto V_{n_0}(t)\prod_{c \in \mathcal{C}}\gamma^{(|c|)}$ changes to $\gamma^{(|c'|+1)}$ for this particular $c'$. The marginal likelihood $m(X^{c'})$ changes to $m(X^{c'} \cup X^{\{i\}})$ for this particular $c'$. Hence, we have the following conditional probability:

$$p(i \to c'|\mathcal{C}_{-i}, \mathcal{D}, \mathbf{T}, \mathbf{T}_0, E) \propto \frac{\gamma^{(|c'|+1)}}{\gamma^{(|c'|)}}\frac{m(X^{c'} \cup X^{\{i\}}|\mathbf{T}, \mathbf{T}_0, E)}{m(X^{c'}|\mathbf{T}, \mathbf{T}_0, E)}$$
$$= (|c'| + \gamma)\frac{m(X^{c'} \cup X^{\{i\}}\}|\mathbf{T}, \mathbf{T}_0, E)}{m(X^{c'}|\mathbf{T}, \mathbf{T}_0, E)}$$

(ii) to a new cluster:

The prior term $V_{n_0}(t)\prod_{c \in \mathcal{C}}\gamma^{(|c|)}$ changes into $V_{n_0}(t+1)\left[\prod_{c \in \mathcal{C}}\gamma^{|c|}\right]\gamma$. Hence, we have the following conditional probability:

$$p(i \to \text{new}|\mathcal{C}_{-i}, \mathcal{D}, \mathbf{T}, \mathbf{T}_0, E)$$
$$\propto \gamma\frac{V_{n_0}(t+1)}{V_{n_0}(t)}m(X^{\{i\}}|\mathbf{T}, \mathbf{T}_0, E)$$

$\square$

**Non-conjugate prior**    When using a non-conjugate prior, we can use Algorithm 8 instead of Algorithm 3. The outline of implementation of Algorithm 8 for FBC can be formulated similar to those of Algorithm 3, as below.

1. Initialize $\mathcal{C} = \{[n_0]\}$ (i.e., a single cluster) with $\phi_{[n_0]} \sim H$.

2. Repeat the following steps $N$ times, to obtain $N$ samples. For $i = 1, \ldots, n_0$: Remove element $i \in [n_0]$ and its matched elements in $J(\{i\})$ from $\mathcal{C}$. Then, generate $m$ independent auxiliary variables $\phi^{(1)}, \ldots, \phi^{(m)} \sim H$. Compute the assignment weights as:

$$w_{c'} = (|c'| + \gamma) \prod_{x \in X^{\{i\}}} f(x|\phi_{c'}), \qquad\qquad c' \in \mathcal{C} \setminus i,$$

$$w_{\text{aux},h'} = \frac{\gamma}{m} \frac{V_{n_0}(t+1)}{V_{n_0}(t)} \prod_{x \in X^{\{i\}}} f(x|\phi^{(h')}), \qquad h' = 1, \ldots, m,$$

where $X^{\{i\}} := \{X_i^{(0)}\} \cup \{X_j^{(1)} : j \in J(\{i\})\}$ and $t$ is a number of clusters when $X^{\{i\}}$ are removed. Then, place them

- to $c' \in \mathcal{C} \setminus i$ with probability $\propto w_{c'}$, or
- to a new randomly chosen cluster $h$ among $m$ auxiliary components, with probability $\propto w_{\text{aux},h}$.

Then, discard all auxiliary variables which are not chosen.

**Proposition B.2.** *The above modification of Algorithm 8 of Neal (2000) is a valid Gibbs sampler.*

*Proof of Proposition B.2.* To justify the proposed modification of Algorithm 8, we only need to calculate the probabilities of replacing $X^{\{i\}}$: (i) to an existing partition $c'$, or (ii) to a new cluster.

The calculation can be done as follows:

(i) to existing $c'$:

The term $\gamma^{(|c'|)}$ in the prior term $p(\mathcal{C}) \propto V_{n_0}(t) \prod_{c \in \mathcal{C}} \gamma^{(|c|)}$ changes to $\gamma^{(|c'|+1)}$ for this particular $c'$. The likelihood is multiplied by $\prod_{x \in X^{\{i\}}} f(x|\phi_{c'})$ for this particular $c'$. Hence, we have the following conditional probability:

$$p(i \to c'|\mathcal{C}_{-i}, \mathcal{D}, \mathbf{T}, \mathbf{T}_0, E) \propto \frac{\gamma^{(|c'|+1)}}{\gamma^{(|c'|)}} \prod_{x \in X^{\{i\}}} f(x|\phi_{c'})$$

$$= (|c'| + \gamma) \prod_{x \in X^{\{i\}}} f(x|\phi_{c'}).$$

(ii) to a new cluster: A new cluster $h$ is chosen uniformly among the auxiliary $m$ components. Then, we have the following conditional probability by Monte-Carlo approximation:

$$p(i \to \text{new}|\mathcal{C}_{-i}, \mathcal{D}, \mathbf{T}, \mathbf{T}_0, E) \propto \gamma \frac{V_{n_0}(t+1)}{V_{n_0}(t)} m(X^{\{i\}}|\mathbf{T}, \mathbf{T}_0, E)$$

$$= \gamma \frac{V_{n_0}(t+1)}{V_{n_0}(t)} \int_{\Theta} \left[ \prod_{x \in X^{\{i\}}} f(x|\phi) \right] H(d\phi)$$

$$\approx \gamma \frac{V_{n_0}(t+1)}{V_{n_0}(t)} \frac{1}{m} \sum_{h'=1}^{m} \prod_{x \in X^{\{i\}}} f(x|\phi^{(h')})$$

$$= \sum_{h'=1}^{m} w_{\text{aux},h'}.$$

From the fact that $h$ is uniformly chosen among $m$-auxiliary components, we have the conditional probability to a new cluster $h$ as follows:

$$p(i \to h|\mathcal{C}_{-i}, \mathcal{D}, \mathbf{T}, \mathbf{T}_0, E) \propto p(i \to h|i \to \text{new}, \cdot) p(i \to \text{new}|\mathcal{C}_{-i}, \mathcal{D}, \mathbf{T}, \mathbf{T}_0, E)$$

$$\propto \frac{w_{\text{aux},h}}{\sum_{h'=1}^{m} w_{\text{aux},h'}} \cdot \sum_{h'=1}^{m} w_{\text{aux},h'}$$

$$= w_{\text{aux},h}.$$

$\square$

**The case when $K$ is known** When to perform FBC with fixed $K$, we only need to consider a point mass prior for $K$. Let $p_K(K) = \mathbb{I}(K = k_*)$ for some fixed $k_*$. Then, we have:

$$V_{n_0}(t) = \sum_{k=1}^{\infty} \frac{k_{(t)}}{(\gamma k)^{(n_0)}} p_K(k) = \frac{(k_*)_{(t)}}{(\gamma k_*)^{(n_0)}} \mathbb{I}(t \leq k_*).$$

For conjugate prior case:

- probability to an existing cluster remains the same.
- probability to a new cluster

$$\propto \gamma \frac{V_{n_0}(t+1)}{V_{n_0}(t)} m(X^{\{i\}} | \mathbf{T}, \mathbf{T}_0, E) = \gamma(k_* - t)_+ m(X^{\{i\}} | \mathbf{T}, \mathbf{T}_0, E).$$

Hence, the probability to a new cluster vanishes when $t$ reaches $k_*$.

For non-conjugate prior case:

- probability to an existing cluster remains the same.
- probability to a new cluster

$$\propto \frac{\gamma}{m} (k_* - t)_+ \prod_{x \in X^{\{i\}}} f(x | \phi^{(h')}).$$

Hence, the probability to a new cluster vanishes when $t$ reaches $k_*$.

**Proposition B.3.** *The above modifications of Algorithm 3, 8 of Neal (2000) is a valid Gibbs sampler for a finite mixture model with $K = k_*$.*

*Proof.* Let $E_i = \{j : Z_j^{(0)} = i\}$, and let $\mathcal{C}(\mathbf{Z})$ be the partition induced by $\mathbf{Z}$. By Dirichlet-multinomial conjugacy, we have:

$$p(\mathbf{Z}) = \int p(\mathbf{Z} | \boldsymbol{\pi}) d\boldsymbol{\pi} = \frac{\Gamma(k_* \gamma)}{\Gamma(\gamma)^{k_*}} \frac{\prod_{i=1}^{n_0} \Gamma(|E_i| + \gamma)}{\Gamma(n_0 + k_* \gamma)} = \frac{1}{(k_* \gamma)^{(n_0)}} \prod_{c \in \mathcal{C}(\mathbf{Z})} \gamma^{(|c|)},$$

for $\mathbf{Z} \in [k_*]^n$.

Therefore, for any partition $\mathcal{C}$ of $[n]$, we have:

$$p(\mathcal{C}) = \sum_{\mathbf{Z} \in [k_*]^{n_0} : \mathcal{C}(\mathbf{Z}) = \mathcal{C}} p(\mathbf{Z})$$

$$= \#\{\mathbf{Z} \in [k_*]^{n_0} : \mathcal{C}(\mathbf{Z}) = \mathcal{C}\} \frac{1}{(\gamma k_*)^{(n_0)}} \prod_{c \in \mathcal{C}} \gamma^{(|c|)}$$

$$= \frac{(k_*)_{(t)}}{(\gamma k_*)^{(n_0)}} \prod_{c \in \mathcal{C}} \gamma^{(|c|)},$$

where $t = |\mathcal{C}|$. $\qquad\square$

## C    EXTENSION OF FBC FOR A MULTINARY SENSITIVE ATTRIBUTE

This section explains that FBC can be modified for the case of a multinary sensitive attribute (i.e., the number of groups $\geq 3$). For simplicity, we consider three sensitive groups. Extension to more than three groups can be done similarly.

Let $\mathcal{D}^{(2)} = \{X_i^{(2)}\}_{i=1}^{n_2}$, along with existing $\mathcal{D}^{(0)}, \mathcal{D}^{(1)}$. Assume that $n_0 \leq \min\{n_1, n_2\}$. Similar to the binary sensitive case, we consider $\mathbf{T}_1 : [n_1] \to [n_0]$ and $\mathbf{T}_2 : [n_2] \to [n_0]$ as random matching maps from $\mathcal{D}^{(1)}$ to $\mathcal{D}^{(0)}$ and from $\mathcal{D}^{(2)}$ to $\mathcal{D}^{(0)}$, respectively. We also consider arbitrary functions $\mathbf{T}_{01} : [n_1] \to [n_0]$ and $\mathbf{T}_{02} : [n_2] \to [n_0]$ and arbitrary subsets $E_1 \in [n_1], E_2 \in [n_2]$ of sizes $m_1$ and $m_2$. Let $\mathcal{C}$ be a partition of $[n_0]$ induced by $\boldsymbol{Z}$ such that $Z_i^{(0)}|\boldsymbol{\pi} \overset{\text{i.i.d.}}{\sim} \text{Categorical}(\cdot|\boldsymbol{\pi}), \forall i \in [n_0]$. Then, similar to Section 3.1, we consider the model

$$
\phi_c \overset{\text{i.i.d.}}{\sim} H, c \in \mathcal{C}
$$

$$
X_i^{(0)} \overset{\text{ind}}{\sim} f(\cdot|\phi_c), i \in c
$$

$$
X_j^{(1)} \overset{\text{ind}}{\sim} \begin{cases} f(\cdot|\phi_c) & \forall j \in [n_1] \setminus E_1 \text{ s.t. } \mathbf{T}_1(j) \in c \\ f(\cdot|\phi_c) & \forall j \in E_1 \text{ s.t. } \mathbf{T}_{01}(j) \in c \end{cases}
$$

$$
X_j^{(2)} \overset{\text{ind}}{\sim} \begin{cases} f(\cdot|\phi_c) & \forall j \in [n_2] \setminus E_2 \text{ s.t. } \mathbf{T}_2(j) \in c \\ f(\cdot|\phi_c) & \forall j \in E_2 \text{ s.t. } \mathbf{T}_{02}(j) \in c \end{cases}
$$

The inference algorithm can be also modified accordingly. Furthermore, $\Delta(\boldsymbol{Z})$ is also generalized as follows. $G$

**Fairness measure for a multinary sensitive attribute**    Let $B$ be the number of sensitive groups. For $b \in \{0, 1, \ldots, B-1\}$, let $n_b$ be the number of samples in group $b$, and denote $\{Z_i^{(b)}\}_{i=1}^{n_b}$ as the cluster assignments of group $b$. Then $\Delta(\boldsymbol{Z})$ is generalized as:

$$
\Delta(\boldsymbol{Z}) := \frac{1}{2(B-1)} \sum_{k=1}^{K} \sum_{b=1}^{B-1} \left| \frac{1}{n_0} \sum_{i=1}^{n_0} \mathbb{I}(Z_i^{(0)} = k) - \frac{1}{n_b} \sum_{j=1}^{n_b} \mathbb{I}(Z_j^{(b)} = k) \right| \in [0, 1]
$$

The Bal is also generalized as: $\texttt{Bal} := \min_{k \in [K]} \texttt{Bal}_k$ where

$$
\texttt{Bal}_k := \min_{b_1 \neq b_2} \left\{ \frac{|C_k^{(b_1)}|}{|C_k^{(b_2)}|}, \frac{|C_k^{(b_2)}|}{|C_k^{(b_1)}|} \right\},
$$

where $C_k^{(b)}$ denotes the set of samples in group $b$ assigned to the $k^{\text{th}}$ cluster. Note that the above formulation of $\Delta(\boldsymbol{Z})$ with $B = 2$ coincides with the definition of $\Delta(\boldsymbol{Z})$ in the binary sensitive case, which is defined in Equation (6). Proposition C.1 below further shows that FBC can control the fairness level $\Delta(\boldsymbol{Z})$ for a multinary sensitive attribute.

**Proposition C.1.** *Denote $n_0$, $n_1$, and $n_2$ as the number of samples in three sensitive groups. Let $\mathbf{T}_1 : [n_1] \to [n_0]$ and $\mathbf{T}_2 : [n_2] \to [n_0]$ be matching maps from groups 1 and 2 to group 0. Consider arbitrary functions $\mathbf{T}_{01} : [n_1] \to [n_0], \mathbf{T}_{02} : [n_2] \to [n_0]$ and arbitrary subsets $E_1 \in [n_1], E_2 \in [n_2]$ of sizes $m_1$ and $m_2$. Suppose that the assignment $\boldsymbol{Z}$ satisfies:*

- *$Z_j^{(1)} = Z_{\mathbf{T}_1(j)}^{(0)}$ for $j \in [n_1] \setminus E_1$ and $Z_j^{(1)} = Z_{\mathbf{T}_{01}(j)}^{(0)}$ for $j \in E_1$,*

- *$Z_j^{(2)} = Z_{\mathbf{T}_2(j)}^{(0)}$ for $j \in [n_2] \setminus E_2$ and $Z_j^{(2)} = Z_{\mathbf{T}_{02}(j)}^{(0)}$ for $j \in E_2$.*

*Then, we have*

$$
\Delta(\boldsymbol{Z}) \leq \frac{1}{2}\left(\frac{m_1}{n_1} + \frac{m_2}{n_2}\right).
$$

*Proof of Proposition C.1.* We investigate all three pairs among the three groups: $(0, 1)$, $(0, 2)$, and $(1, 2)$.

(i) Between groups 0 and 1: Since $Z_j^{(1)} = Z_{\mathbf{T}_1(j)}^{(0)}$ for $j \in [n_1] \setminus E_1$, we have the following inequality utilizing the proof of Proposition A.4,

$$\frac{1}{2} \sum_{k=1}^{K} \left| \frac{1}{n_0} \sum_{i=1}^{n_0} \mathbb{I}(Z_i^{(0)} = k) - \frac{1}{n_1} \sum_{j=1}^{n_1} \mathbb{I}(Z_j^{(1)} = k) \right| \leq \frac{m_1}{n_1}.$$

(ii) Between groups 0 and 2: Similarly, we have

$$\frac{1}{2} \sum_{k=1}^{K} \left| \frac{1}{n_0} \sum_{i=1}^{n_0} \mathbb{I}(Z_i^{(0)} = k) - \frac{1}{n_2} \sum_{j=1}^{n_2} \mathbb{I}(Z_j^{(2)} = k) \right| \leq \frac{m_2}{n_2}.$$

Combining the two terms: Taking the average, we get:

$$\Delta(\boldsymbol{Z}) = \frac{2}{2(3-1)} \sum_{b=1}^{2} \left( \frac{1}{2} \sum_{k=1}^{K} \left| \frac{1}{n_0} \sum_{i=1}^{n_0} \mathbb{I}(Z_i^{(0)} = k) - \frac{1}{n_b} \sum_{j=1}^{n_b} \mathbb{I}(Z_j^{(b)} = k) \right| \right)$$

$$\leq \frac{1}{2} \left( \frac{m_1}{n_1} + \frac{m_2}{n_2} \right).$$

$\square$

Note that we use this extended approach for BANK dataset with three sensitive groups in our experiments. See Section E.3 for the results showing that FBC works well for a multinary sensitive attribute.

# D  HIERARCHICAL PRIOR ON $K$ WHEN $K$ IS UNKNOWN

Here, we discuss how to consider a hierarchical prior on $K$, to make the sampling of $K$ robust. A simple idea is to consider a Beta prior on $\kappa$, where the prior of $K$ is Geometric($\kappa$). Then, we can choose one from the following two options when considering posterior inference with varying $\kappa$. For simplicity, we utilize the $\kappa$-marginalized version in our experiments.

**Marginalize $\kappa$**   One simple idea is to marginalize $\kappa$, and consider the corresponding distribution of $K$.

$$\kappa \sim \text{Beta}(a, b)$$
$$K|\kappa \sim \text{Geometric}(\kappa)$$

We can marginalize the above hierarchy as follows:

$$\bar{p}_K(k) = \int p_K(k|\kappa)p(\kappa)d\kappa$$
$$= \int_0^1 \kappa(1-\kappa)^{k-1}\frac{\kappa^{a-1}(1-\kappa)^{b-1}}{B(a,b)}d\kappa$$
$$= \frac{B(a+1, b+k-1)}{B(a,b)}.$$

Approximately for large $k$, $\bar{p}_K(k) \sim k^{-(a+1)}$, which makes $\mathbb{E}[K] < \infty$ when $a > 1$. Considering a uniform prior for $\kappa$ is equivalent with Beta$(1, 1)$, resulting in $\mathbb{E}[K] = \infty$.

Note that, this is a simple change of the prior of $K$, which does not affect the procedure of FBC at all. That is, we just alter the term $p_K$ as $\bar{p}_K$.

**Sample $\kappa$**   An alternative is to sample $\kappa$. To do so, we need to investigate whether the sampling of $\kappa$ can be well integrated to the current formulation of FBC. We have:

$$p(\kappa|K) \propto \kappa^{a-1}(1-\kappa)^{b-1} \times \kappa(1-\kappa)^{K-1} = \kappa^a(1-\kappa)^{b+K-2} \sim \text{Beta}(a+1, b+K-1)$$

Then, the posterior of $K$ given partition $\mathcal{C}$ can be yielded as:

$$p(K|\mathcal{C}, \kappa) \propto \kappa(1-\kappa)^{K-1}\frac{(K)_t}{(\gamma K)^{(n)}}$$

Hence, we can define a similar term to $V_n(t)$ as:

$$V_n^{(\kappa)}(t) := \sum_{k=1}^\infty \kappa(1-\kappa)^{K-1}\frac{(k)_t}{(\gamma k)^{(n)}}$$

Finally, we get

$$p(\mathcal{C}|\kappa) = V_n^{(\kappa)}(t)\prod_{c\in\mathcal{C}}\gamma^{(|c|)}$$

We here can simply use a Gibbs sampler for $\kappa$ as well as other parameters. That is, we update $\kappa$ from:

$$p(\kappa|\mathcal{C}) \propto p(\kappa)V_n^{(\kappa)}(t)$$

Note that this update requires a proper sampling algorithm such as the MH algorithm.

- Proposal: $\kappa' \sim \text{Beta}(a', b')$

# E  EXPERIMENTS

## E.1  DATASETS

1. Toy dataset: We build a 2D toy dataset from a 6-component Gaussian mixture model with unit covariance matrix $\mathbb{I}_2$. For $\mathcal{D}^{(0)}$, we draw 600 samples from each $\mathcal{N}([-5, -30], \mathbb{I}_2)/3 + \mathcal{N}([-5, 0], \mathbb{I}_2)/3 + \mathcal{N}([-5, 30], \mathbb{I}_2)/3$ and Similarly for $\mathcal{D}^{(1)}$, we draw 600 samples from $\mathcal{N}([-5, -29.5], \mathbb{I}_2)/3 + \mathcal{N}([-5, 0.5], \mathbb{I}_2)/3 + \mathcal{N}([-5, 30.5], \mathbb{I}_2)/3$. As a result, the total number of samples is 1200, with $n_0 = n_1 = 600$.

2. ADULT: The adult income dataset is a collection of data consisting of several demographic features including employment features. It is extracted from 1994 U.S. Census database (Becker & Kohavi, 1996). Sensitive group sizes are (10,771 / 21,790). We use 5 continuous features (age, fnlwgt, education-num, capital-gain, hours-per-week). For the sensitive attribute, we use the gender (male/female) attribute.

3. BANK: The bank marketing dataset is a collection of data from a Portuguese bank's direct marketing campaigns, each corresponding to an individual client contacted (Moro et al., 2014). We use 6 continuous features (age, call duration, 3-month Euribor rate, number of employees, consumer price index, and number of contacts during the campaign). For two sensitive groups, we treat marital status as the sensitive attribute: categorized into two groups (single/married) and exclude all 'unknown' entries. Sensitive group sizes are (16,180 / 24,928).

   When considering three sensitive groups, following (Ziko et al., 2021), we categorize the marital status into three groups (single/married/divorced) and exclude all 'unknown' entries. Sensitive group sizes are (4,612 / 11,568 / 24,928).

4. DIABETES: The diabetes dataset is a collection of data spanning five years, consisting of various physical indicators (e.g., glucose concentration, blood pressure, BMI, etc., totaling 7 features) of Pima Indian women[1]. Sensitive group sizes are (372 / 396). It originates from the National Institute of Diabetes and Digestive and Kidney Diseases (Smith et al., 1988). For the sensitive attribute, we use the binarized age attribute at the median value.

## E.2  IMPLEMENTATION DETAILS

**Algorithms**

- Baseline methods: For MFM, we employ the Julia code of Miller & Harrison (2018) without modification, available on the authors' GitHub[2]. Similarly, for SFC, VFC, FCA and FairDen, we use the publicly released source codes provided by the authors[3][4][5][6].

- FBC: We use a conjugate prior for $H$. For Gaussian mixture, we use the following conjugate prior following Miller & Harrison (2018). For $\boldsymbol{\theta}$, we set $a = b = 1$ in $\mathcal{N}(\mu_j, \lambda_j^{-1})$, $\lambda \sim$ Gamma$(a, b)$, $\mu_j | \lambda \sim \mathcal{N}(0, \lambda^{-1})$. For $\mathbf{T}$, we set the temperature $\tau = 1.0$. For fixed $K$ cases, we simply set $\gamma = 1$ for the Dirichlet distribution. We repeat the step 1 for $(1, 10, 5)$ for $k_* = 10$ (for ADULT, BANK and DIABETES, respectively). We use 6000 burn-in epochs and select a sample from the posterior mode among 4000 samples after burn-in for ADULT and BANK, and we use 2000 burn-in epochs and select a sample from 2000 samples after burn-in for DIABETES.

  For unknown $K$ cases, we consider $\kappa \sim$ Beta$(21, 80)$ and $\gamma = (10.0, 10.0, 5.0)$ for the Dirichlet distribution to prevent the cluster numbers explode (for ADULT, BANK and DIABETES, respectively). We use 8000 burn-in epochs and select a sample from the posterior mode among 2000 samples after burn-in for ADULT, 18000 burn-in epochs and select a sample from 2000 samples after burn-in for BANK, and we use 8000 burn-in epochs and

---

[1]Diabetes: https://github.com/aasu14/Diabetes-Data-Set-UCI
[2]MFM: https://github.com/jwmi/BayesianMixtures.jl
[3]SFC: https://github.com/talwagner/fair_clustering
[4]VFC: https://github.com/imtiazziko/Variational-Fair-Clustering
[5]FCA: https://github.com/kwkimonline/FCA
[6]FairDen: https://jugit.fz-juelich.de/ias-8/fairden

select a sample from 2000 samples after burn-in for DIABETES. The choice is because, when $K$ is treated as a random variable, since the parameter space with unknown $K$ is much larger than that with fixed $K$. We also utilize the Jain-Neal split-merge sampler (Jain & Neal, 2004) to enable faster mixing, following the setting of Miller & Harrison (2018).

For the choice of $D$ in the energy, we use the Euclidean distance for continuous features and the Gower distance (Gower, 1971) for mixed-type data (i.e., continuous + categorical) which is a weighted sum of scaled euclidean distance and hamming distance (Hamming, 1950; Huang, 1998; Zhang et al., 2006).

For faster convergence, we partially initialize $\mathbf{T}$ utilizing optimal transport, motivated by prior work showing that matching nearby instances tends to yield higher clustering utility (Chierichetti et al., 2017; Backurs et al., 2019; Kim et al., 2025b). In detail, we randomly sample some proportion from each (upsampled) $\mathcal{D}^{(s)}$ and calculate the optimal transport between them, while remains are randomly matched.

**Performance measures**

- `NLD`: For the baseline methods, `NLD` is computed as follows: we first estimate a mean for each cluster and a single variance shared across all clusters/components, then compute the negative log-likelihood of the data under this fitted mixture model.

- `Acc`: Even when the number of clusters differs from the number of class labels, we assign to each cluster the ground-truth class that appears most frequently within that cluster (majority vote). We then compute `Acc` as the fraction of samples whose predicted labels (from this assignment) match their ground-truth class labels.

**Hardwares**

- The Julia language is used for running FBC.

- All our experiments are done through Julia 1.11.2, Python 3.9.16 with Intel(R) Xeon(R) Silver 4310 CPU @ 2.10GHz and 128GB RAM.

### E.3 EXPERIMENTAL RESULTS

#### E.3.1 FAIR CLUSTERING PERFORMANCE (KNOWN $K$)

**Comparison of FBC to existing fair clustering algorithms**   Table 2 compares the computation time of FBC and FCA, which suggests that FBC requires less computation time than FCA, up to 50% faster.

Table 2: Averaged computation time (seconds) of FBC and FCA on five random trials for $k_* = 10$.

| Computation time | ADULT | BANK | DIABETES |
|---|---|---|---|
| FCA | 1059.22 | 1365.32 | 26.71 |
| FBC ✓ | **599.69** | **812.08** | **12.38** |

Similar to Table 1 in the main body, we investigate whether FBC yields reasonable clustering results compared to baselines when $K$ is fixed at $k_* = 2$. We similarly run each algorithm to achieve the maximum fairness (e.g., $\Delta \approx 0$) for a fair comparison. Table 3 shows the results, suggesting that FBC is competitive to the baselines when $K$ is fixed at $k_* = 2$.

Table 3: Comparison of utility (Cost, NLD, Acc) and fairness levels ($\Delta$, Bal) for $k_* = 2$. Bold-faced values and underlined values (Cost and $\Delta$) indicate the best and second-best values, respectively, among the methods that achieve near-perfect fairness levels.

| Dataset | Measure | | SFC | VFC | FCA | FBC ✓ |
|---|---|---|---|---|---|---|
| ADULT | UTILITY | Cost ($\downarrow$) | 4.397 | 4.248 | **4.237** | 4.259 |
| | | NLD ($\uparrow$) | 6.684 | 7.129 | 6.658 | 6.688 |
| | | Acc ($\uparrow$) | 0.763 | 0.636 | 0.765 | 0.766 |
| | FAIRNESS | $\Delta$ ($\downarrow$) | **0.000** | 0.041 | **0.000** | **0.000** |
| | | Bal ($\uparrow$) | 0.494 | 0.455 | 0.494 | 0.494 |
| BANK | UTILITY | Cost ($\downarrow$) | 4.042 | 3.729 | **3.919** | **3.919** |
| | | NLD ($\uparrow$) | 7.739 | 7.717 | 7.746 | 7.758 |
| | | Acc ($\uparrow$) | 0.733 | 0.720 | 0.680 | 0.681 |
| | FAIRNESS | $\Delta$ ($\downarrow$) | 0.003 | 0.084 | **0.000** | 0.002 |
| | | Bal ($\uparrow$) | 0.647 | 0.571 | 0.649 | 0.647 |
| DIABETES | UTILITY | Cost ($\downarrow$) | 6.621 | 5.729 | **5.733** | 5.958 |
| | | NLD ($\uparrow$) | 9.838 | 9.752 | 9.753 | 9.707 |
| | | Acc ($\uparrow$) | 0.613 | 0.638 | 0.633 | 0.697 |
| | FAIRNESS | $\Delta$ ($\downarrow$) | 0.017 | 0.002 | 0.001 | **0.000** |
| | | Bal ($\uparrow$) | 0.884 | 0.937 | 0.937 | 0.939 |

**Fairness level control of FBC** Figure 8 shows the relationship between $m$ and the fairness level, showing that fairness level is well controlled by controlling $m$ unless $m$ is too large.

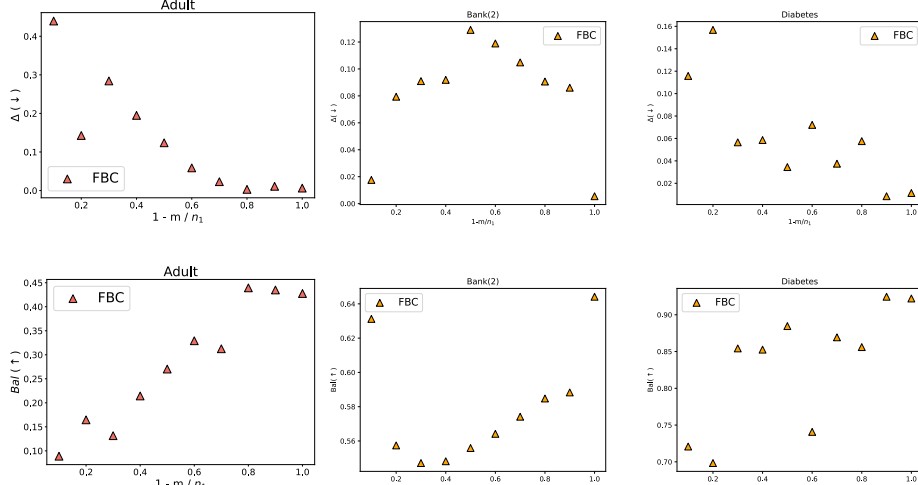

Figure 8: (Top three) Trade-off between $m$ (the size of $E$) and the fairness level $\Delta$. Smaller $\Delta$, fairer the clustering. (Bottom three) Trade-off between $m$ (the size of $E$) and the `Bal`. Larger `Bal`, fairer the clustering.

**Additional analysis: handling a multinary sensitive attribute (BANK)** AWe analyze BANK with three sensitive groups. Table 4 presents the performance comparison between VFC and FBC, showing that FBC is competitive to VFC in terms of the utility–fairness trade-off. In particular, FBC achieves better fairness levels (i.e., lower $\Delta$ and higher `Bal`), while its utility (`Cost` and `Acc`) remains comparable.

Table 4: Utility (`Cost`, `Acc`) and fairness ($\Delta$, `Bal`) on BANK for $k_* = 10$.

| $K$ | Measure | | VFC | FBC ✓ |
|---|---|---|---|---|
| 10 | UTILITY | `Cost` ($\downarrow$) | 1.578 | 2.310 |
| | | `Acc` ($\uparrow$) | 0.742 | 0.891 |
| | FAIRNESS | $\Delta$ ($\downarrow$) | 0.079 | 0.002 |
| | | `Bal` ($\uparrow$) | 0.166 | 0.183 |

### E.3.2    REASONABLE INFERENCE OF $K$

**Cluster quality**    Figure 9 draws the posterior distributions of $K$ for the three datasets. The inferred $K$s for the three datasets are sampled from the posterior modes, and posterior distributions are well-concentrated around the posterior modes.

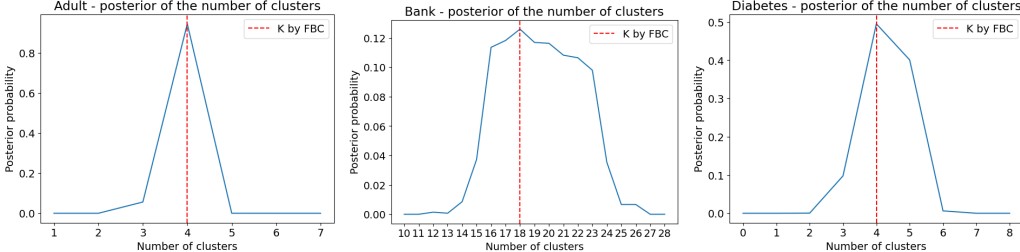

Figure 9: Posteriors of $K$ on (left) ADULT, (center) BANK, and (right) DIABETES datasets.

Table 5 below shows that FBC still performs comparable to baseline methods when $K$ is unknown so inferred, in terms of fairness-utility trade-off.

Table 5: Utility (Cost, Acc) and fairness ($\Delta$, Bal) on three datasets with inferred $K$s. Bold-faced values and underlined values (Cost and $\Delta$) indicate the best and second-best values, respectively, among the methods that achieve near-perfect fairness levels.

| Dataset | Measure | | SFC | VFC | FCA | FBC ✓ |
|---|---|---|---|---|---|---|
| ADULT ($K = 4$) | UTILITY | Cost ($\downarrow$) | 4.045 | 3.317 | **3.033** | 4.364 |
| | | Acc ($\uparrow$) | 0.763 | 0.759 | 0.766 | 0.761 |
| | FAIRNESS | $\Delta$ ($\downarrow$) | 0.001 | 0.031 | **0.000** | **0.000** |
| | | Bal ($\uparrow$) | 0.491 | 0.443 | 0.494 | 0.494 |
| BANK ($K = 18$) | UTILITY | Cost ($\downarrow$) | 2.570 | 1.112 | **1.377** | 2.202 |
| | | Acc ($\uparrow$) | 0.893 | 0.892 | 0.892 | 0.891 |
| | FAIRNESS | $\Delta$ ($\downarrow$) | 0.003 | 0.063 | **0.001** | 0.006 |
| | | Bal ($\uparrow$) | 0.647 | 0.466 | 0.644 | 0.612 |
| DIABETES ($K = 4$) | UTILITY | Cost ($\downarrow$) | 5.771 | 4.858 | **4.646** | 5.381 |
| | | Acc ($\uparrow$) | 0.656 | 0.656 | 0.702 | 0.682 |
| | FAIRNESS | $\Delta$ ($\downarrow$) | 0.007 | 0.112 | **0.001** | 0.006 |
| | | Bal ($\uparrow$) | 0.824 | 0.671 | 0.938 | 0.875 |

**Density estimation**    We further support the optimality of $K$ inferred by FBC, in terms of density estimation. To this end, we first split a given dataset into training/test with ratios 8:2, then run FBC on the training dataset and compute the posterior density on the test dataset. The results are given in Section 5.2 in the main body.

### E.3.3 FLEXIBILITY OF FBC: APPLICABILITY TO VARIOUS DATA TYPES

FBC can be applied to any variable type provided that the likelihood is defined. In this analysis, we consider datasets which include both continuous variables and categorical variables.

**Dataset construction** For a baseline method, we consider FairDen, since it can also applied to both continuous or categorical variables. Note that we use subsampled ADULT dataset in this analysis, since FairDen requires high computational complexity when using the entire ADULT. In detail, it requires building a full $n \times n$ similarity matrix between data points, calculating the Laplacian, and performing eigen-decomposition under fairness constraints. Thus the computational complexity of this pipeline is $\mathcal{O}(n^2)$ in both time and memory, and involves fragile linear algebra, leading to numerical instability as well as computational overhead when using the entire ADULT dataset.

That is, we randomly select $2,000$ subsamples from the entire dataset. Note that the resulting subsample contains $1,357$ males and $643$ females, so the maximum achievable value of `Bal` is $643/1357 \approx 0.474$.

In addition to continuous features used in the main analysis, we consider 2 categorical features (marital status and race with 7 and 5 categories, respectively).

**FBC for mixed-type (continuous + categorical) data** For continuous variables, we keep considering the Gaussian mixture. For categorical variables, we consider the mixture of independent Multinoulli distributions, where each component $f(\cdot|\theta_k)$ is the product of Multinoulli distribution. In other words, $f(\cdot|\theta_k) \sim \prod_{j=1}^{d_{\text{cate}}} \text{Cat}(\cdot; \phi_{k,j})$, where $\phi_{k,j} = (p_{k,j,1}, \ldots, p_{k,j,l_j}) \in [0,1]^{l_j}$. Here, $l_j$ is the number of categories of the $j$-th categorical feature. We use the conjugate prior as $\phi_{k,j} \sim \text{Dirichlet}(\alpha, \ldots, \alpha)$ with $\alpha = 1$.

**Results** The comparison results are provided in Table 6. For both FairDen and FBC, the additional use of categorical variables improves `Acc`, compared to the results with the continuous variables only. Note that we cannot measure `Cost`, which cannot be applied to categorical variables. However, while the use of categorical variables reduces `Bal` and increases $\Delta$ in FairDen, FBC still achieves nearly perfect fairness in terms of $\Delta$ and `Bal` for both cases, with and without categorical variables. In summary, FBC outperforms FairDen for both cases, with and without categorical variables.

Table 6: Comparison of utility (`Acc`) and fairness ($\Delta$, `Bal`) on subsampled ADULT for $k_* = 10$. **Bold**-faced results indicate the bests. 'c' denotes for the results with the use of continuous variables only, and 'cc' denotes for the results with the use of both continuous and categorical variables.

| Dataset | Measure | | FairDen(c) | FairDen(cc) | FBC(c) ✓ | FBC(cc) ✓ |
|---|---|---|---|---|---|---|
| ADULT(SUB) | UTILITY | Acc (↑) | 0.651 | 0.654 | 0.715 | **0.721** |
| | FAIRNESS | Δ (↓) | 0.005 | 0.013 | 0.001 | **0.001** |
| | | Bal (↑) | 0.466 | 0.451 | 0.472 | **0.473** |

### E.3.4 FLEXIBILITY OF FBC: CLUSTERING UNDER SIZE CONSTRAINTS

**FBC algorithm under the cluster size constraint** Consider a problem of searching fair clusters under the constraint that the maximum size of each cluster should be bounded by $M$. For this purpose, we can modify Algorithm 3 for FBC under the size constraint as:

1. Initialize $\mathcal{C} = \{[n_0]\}$ (i.e., a single cluster)

2. Repeat the following steps $N$ times, to obtain $N$ samples. For $i = 1, \ldots, n_0$: Remove element $i \in [n_0]$ and its matched elements in $J(\{i\}; \mathbf{T}, \mathbf{T}_0, E) := \{j \in E : \mathbf{T}_0(j) = i\} \cup \{j \in [n_1] \setminus E : \mathbf{T}(j) = i\}$ from $\mathcal{C}$. Then, place them
   - to $c' \in \mathcal{C} \setminus i$ with probability

$$\propto (|c'| + \gamma) \frac{m(X^{c'} \cup X^{\{i\}} | \mathbf{T}, \mathbf{T}_0, E)}{m(X^{c'} | \mathbf{T}, \mathbf{T}_0, E)} \mathbb{I}(|c'| \leq M)$$

   where $X^{\{i\}} := \{X_i^{(0)}\} \cup \{X_j^{(1)} : j \in J(\{i\})\}$.
   - to a new cluster with probability $\propto \gamma \frac{V_{n_0}(t+1)}{V_{n_0}(t)} m(X^{\{i\}} | \mathbf{T}, \mathbf{T}_0, E)$, where $t$ is a number of clusters when $X^{\{i\}}$ are removed.

The modification of Algorithm 8 for FBC can be done similarly.

**Experimental setup and Baseline method** Given $\alpha \geq 1$, we set the per-cluster upper bound $U_{\max}^{(\alpha)}(K) = \lceil \alpha \frac{n}{K} \rceil$. That is, $\alpha = 1$ regularizes the sizes of all clusters to the uniform size of $n/K$, and larger $\alpha$ allows more relaxations.

As a baseline, we consider a post-processing fair clustering method proposed by (Bera et al., 2019), which aims to find fair assignments through solving a linear program with fixed pre-determined cluster centers. Since the size constraint is also linear with respect to the assignments, we add it with the parameter $\alpha$, when solving the linear program.

**Results** We consider varying $\alpha \in \{1.25, 1.50, 1.75\}$, whose corresponding cluster size upper bounds are $\{4071, 4885, 5699\}$. Results under these size constraints are reported in Table 7. FBC consistently outperforms the post-processing method of Bera et al. (2019). Specifically, for Bera et al. (2019), the constraint does not affect the fair assignment when $\alpha$ exceeds 1.5; in that case, the optimal Cost for Bera et al. (2019) is 2.748 - higher than that of FBC. These findings demonstrate the superiority of FBC for strictly size-constrained fair clustering.

Various size constrained clustering algorithms have been considered (Esmaeili et al., 2020; Zhu et al., 2010; Höppner & Klawonn, 2008). We believe that FBC could be modified for such problems without much hamper similarly to what we have done for the upper bound constraint.

Table 7: Results of utility (Cost), fairness ($\Delta$, Bal), and statistics on cluster sizes (min size, max size) on ADULT with $k_* = 10$ under size constraints. $\alpha$ controls the size constraint.

| $\alpha$ | Measure | | Bera et al. (2019) under size constraint with $\alpha$ | FBC ✓ |
|---|---|---|---|---|
| | UTILITY | Cost ($\downarrow$) | 2.829 | 2.481 |
| 1.25 | FAIRNESS | $\Delta$ ($\downarrow$) | 0.002 | 0.001 |
| | | Bal ($\uparrow$) | 0.492 | 0.493 |
| | SIZES | min / max | 2,802 / 4,071 | 219 / 4,070 |
| | UTILITY | Cost ($\downarrow$) | 2.748 | 2.451 |
| 1.50 | FAIRNESS | $\Delta$ ($\downarrow$) | 0.000 | 0.001 |
| | | Bal ($\uparrow$) | 0.494 | 0.485 |
| | SIZES | min / max | 2,802 / 4,128 | 98 / 4,885 |
| | UTILITY | Cost ($\downarrow$) | 2.748 | 2.439 |
| 1.75 | FAIRNESS | $\Delta$ ($\downarrow$) | 0.000 | 0.001 |
| | | Bal ($\uparrow$) | 0.494 | 0.491 |
| | SIZES | min / max | 2,802 / 4,128 | 79 / 5,699 |

### E.3.5 ABLATION STUDIES

**Temperature** $\tau$   We analyze the impact of the temperature constant $\tau$ in the prior of $\mathbf{T}$ defined in Definition 3.1. To do so, we set $E = \emptyset$ and compare the utility (Cost) and fairness levels (Bal and $\Delta$) for various values of $\tau$. Table 8 below reports the performance of FBC for different temperature values $\tau \in \{0.1, 1.0, 10.0\}$. Overall, varying $\tau$ does not affect much to the performance of FBC.

Table 8: Comparison of $K$, Cost, Acc, $\Delta$, and Bal for $\tau \in \{0.1, 1.0, 10.0\}$.

| Dataset | Measure | $\tau$ | | |
|---|---|---|---|---|
| | | 0.1 | 1.0 | 10.0 |
| ADULT | $K$ | 4 | 4 | 4 |
| | Cost ($\downarrow$) | 4.364 | 4.364 | 4.364 |
| | Acc ($\uparrow$) | 0.761 | 0.761 | 0.761 |
| | $\Delta$ ($\downarrow$) | 0.000 | 0.000 | 0.000 |
| | Bal ($\uparrow$) | 0.494 | 0.494 | 0.494 |
| BANK | $K$ | 18 | 18 | 18 |
| | Cost ($\downarrow$) | 2.202 | 2.202 | 2.202 |
| | Acc ($\uparrow$) | 0.891 | 0.891 | 0.891 |
| | $\Delta$ ($\downarrow$) | 0.006 | 0.006 | 0.006 |
| | Bal ($\uparrow$) | 0.612 | 0.612 | 0.612 |
| DIABETES | $K$ | 4 | 4 | 4 |
| | Cost ($\downarrow$) | 5.381 | 5.381 | 5.381 |
| | Acc ($\uparrow$) | 0.682 | 0.682 | 0.682 |
| | $\Delta$ ($\downarrow$) | 0.006 | 0.006 | 0.006 |
| | Bal ($\uparrow$) | 0.875 | 0.875 | 0.875 |

**Choice of $R$ when $r > 0$**   In this section, we compare the two choices of $R$ in Section 2.1.2: (i) a random subset of $[n_0]$ and (ii) the index of samples closest to the cluster centers obtained by a certain clustering algorithm to $\mathcal{D}^{(0)}$ with $K = r$, which is considered in Section 2.1.2. Table 9 below provides the results, showing that the performance of the two approaches are not much different. Here, we utilized $K$-medoids algorithm to yield the cluster centers from $[n_0]$. Overall, we can conclude that FBC is not sensitive to the choice of $R$ and the two proposed heuristic approaches work well in practice.

Table 9: Comparison of $K$, Cost, Acc, $\Delta$, and Bal for the two heuristic approaches in Section 2.1.2. 'Random' indicates the first approach ($R =$ a random subset of $[n_0]$). and 'Clustering' indicates the second approach ($R =$ centers obtained by a clustering algorithm).

| Dataset | Measure | $R$ | |
|---|---|---|---|
| | | Random | Clustering |
| ADULT | $K$ | 4 | 4 |
| | Cost ($\downarrow$) | 4.364 | 4.364 |
| | Acc ($\uparrow$) | 0.761 | 0.761 |
| | $\Delta$ ($\downarrow$) | 0.000 | 0.000 |
| | Bal ($\uparrow$) | 0.494 | 0.494 |
| BANK | $K$ | 18 | 18 |
| | Cost ($\downarrow$) | 2.202 | 2.202 |
| | Acc ($\uparrow$) | 0.891 | 0.891 |
| | $\Delta$ ($\downarrow$) | 0.006 | 0.006 |
| | Bal ($\uparrow$) | 0.612 | 0.612 |
| DIABETES | $K$ | 4 | 4 |
| | Cost ($\downarrow$) | 5.381 | 5.381 |
| | Acc ($\uparrow$) | 0.682 | 0.682 |
| | $\Delta$ ($\downarrow$) | 0.006 | 0.006 |
| | Bal ($\uparrow$) | 0.875 | 0.875 |

**Variation of covariance structure**  We examine how the covariance structure in the Gaussian mixture model affects utility and fairness of FBC. Specifically, we compare the *Unified* covariance used in the main analysis (a single scale parameter $\lambda$ shared across clusters and features) with a more flexible one, *Iso*tropic covariance (cluster-wise $\lambda_j$). To do so, we split the dataset into training/test with 8:2 ratios, and calculate the performance on the test data. We also fix $K = k_* = 10$ in this analysis.

The results in Table 10 give the following implications: (i) allowing cluster-specific covariances improves density estimation i.e., lowers `NLD`, by capturing richer covariance variability; (ii) *Unified* offers lower `Cost`, since `Cost` assumes shared variances across clusters; (iii) fairness is achieved well regardless of the covariance choice: both settings maintain near-perfect fairness ($\Delta \approx 0$ and high `Bal`).

Table 10: Comparison of utility (`Cost`, `NLD`, `Acc`) and fairness ($\Delta$, `Bal`). $K$ is fixed as $K = k_* = 10$. *Unified* indicates the use of a single scale parameter and *Iso* indicates the use of cluster-specific scale parameters.

| Dataset | Measure | Covariance | |
|---|---|---|---|
| | | *Unified* | *Iso* |
| ADULT | Cost ($\downarrow$) | 1.954 | 2.272 |
| | NLD ($\downarrow$) | 6.242 | 6.239 |
| | Acc ($\uparrow$) | 0.781 | 0.775 |
| | $\Delta$ ($\downarrow$) | 0.001 | 0.001 |
| | Bal ($\uparrow$) | 0.491 | 0.492 |
| BANK | Cost ($\downarrow$) | 1.787 | 1.788 |
| | NLD ($\downarrow$) | 6.302 | 6.300 |
| | Acc ($\uparrow$) | 0.890 | 0.888 |
| | $\Delta$ ($\downarrow$) | 0.004 | 0.004 |
| | Bal ($\uparrow$) | 0.615 | 0.614 |
| DIABETES | Cost ($\downarrow$) | 3.757 | 3.958 |
| | NLD ($\downarrow$) | 9.088 | 9.076 |
| | Acc ($\uparrow$) | 0.693 | 0.701 |
| | $\Delta$ ($\downarrow$) | 0.013 | 0.018 |
| | Bal ($\uparrow$) | 0.882 | 0.800 |

## E.4 CONVERGENCE OF MCMC

We assess the convergence of the MCMC algorithm by monitoring (i) the autocorrelation function of the inferred $K$ and (ii) the negative log-likelihood (NLL) on training data (i.e., the observed instances), over sampling iterations. As is done by Miller & Harrison (2018), the results in Figures 10 and 11 of Section E.3.5 show that the MCMC algorithm converges well and finds good clusters.

$K$: As is done by Miller & Harrison (2018), in Figure 10, we draw the autocorrelation function $\rho(h)$ defined as
$$\rho(h) = \text{corr}\{(K_t, K_{t+h}), t = 1, \ldots\},$$
where $K_t$ is a posterior sample at iteration $t$. The autocorrelation functions amply support that the proposed MCMC algorithm converges well.

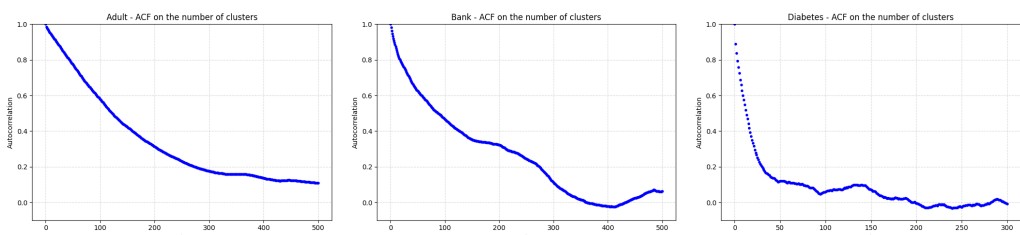

Figure 10: $h$ vs. Autocorrelation functions for (left) DIABETES, (center) ADULT, and (right) BANK datasets.

NLL: Figure 11 draws the trace plots of the NLL on training data. Dramatic decreases of NLL are observed which would happen when the MCMC algorithm moves one local optimum to another local optimum. This supports the ability of FBC to explore high-posterior clusters efficiently.

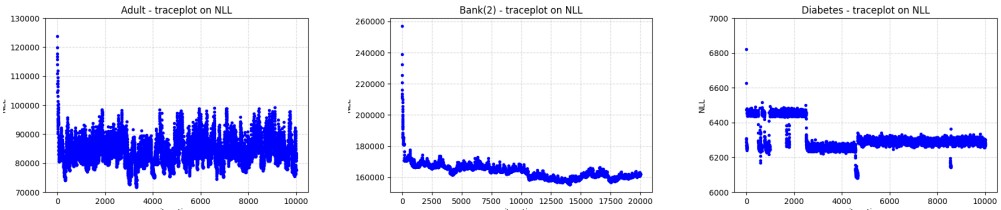

Figure 11: Trace plots of NLL on (left) DIABETES, (center) ADULT, and (right) BANK datasets.

# F ADDITIONAL EXPERIMENTS

## F.1 ASSIGNMENTS FOR NEW DATA

Suppose the new dataset is given as $D_0^{\text{new}} \cup D_1^{\text{new}}$. Then, we assign instances of $D_0^{\text{new}}$ to their nearest neighbors in $D_0$, and similarly assign instances of $D_1^{\text{new}}$ to their nearest neighbors in $D_1$.

We empirically validate this assignment strategy as follows: (i) we split the ADULT dataset into training and test sets with an 8:2 ratio; (ii) we perform FBC on the training data and then assign the test data according to the procedure described above.

The results are reported in Table 11, where we observe that fairness on the new (test) data (in terms of $\Delta$ and Balance) remains near the values of near-perfect fairness, and the NLD value is similar to that in Table 1, where the full dataset is used. These empirical findings suggest that the proposed assignment technique for new data performs well in practice.

Table 11: Performance comparison between FBC performed/evaluated on the full data and FBC performed on the training data and evaluated on the test data, on ADULT dataset.

|  | NLD ($\downarrow$) | $\Delta$ ($\downarrow$) | Bal ($\uparrow$) |
|---|---|---|---|
| Full data | 6.242 | 0.001 | 0.491 |
| New data | 6.241 | 0.019 | 0.472 |

## F.2 COMPUTATIONAL COST

The computational complexity of the MH step (STEP 1) for $\mathbf{T}$ does not depend on the input dimension, and its scale on each iteration is $O(1)$ since it is a random swap with a fixed number of repetition (10 times in our experiments). The computational complexity of the mixture parameter sampling of each iteration (STEP 2) for $\Phi$, is empirically observed as $O(n)$ in Miller & Harrison (2018).

To further verify the scalability of FBC, we compare its runtime with baseline methods for several datasets. For this purpose, we plot the logarithm of dataset size ($n$) vs. runtime in Figure 12 (datasets are DIABETES, ADULT, BANK, and CENSUS). The results show that FBC is competitive to baseline methods in terms of computational cost, while maintaining the nearly-best performance.

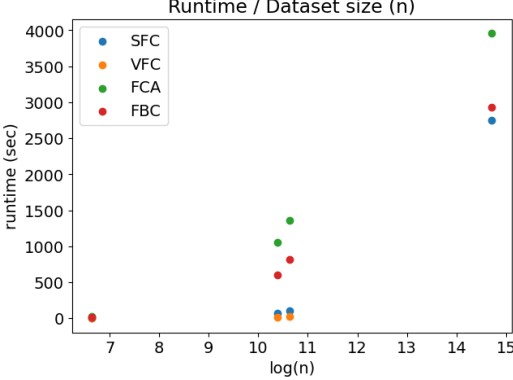

Figure 12: Scatterplot of runtime and the logarithm of dataset size.

## F.3 PERFORMANCE ON A LARGE-SCALE DATASET

**Known $K$** To validate the scalability of FBC, we conduct an additional experiment using Census dataset, which has a scale of millions and has been previously used in the fairness literature (Backurs et al., 2019; Ziko et al., 2021). It [7] is a sub-dataset of the 1990 US Census, consisting of 2,458,285

---

[7] https://archive.ics.uci.edu/dataset/116/us+census+data+1990

samples with 68 attributes. We use 25 continuous variable and consider 'gender' attribute as the sensitive attribute, similar to the approach of Backurs et al. (2019) and Ziko et al. (2021). Since it is noted that for CENSUS dataset, VFC fails without $L_2$ normalization (Kim et al., 2025b), we only compare FBC to SFC and FCA.

In Table 12, we can observe that the similar behavior to that of Table 1. While all methods achieves near-perfect fairness, FBC yields better `Cost` than SFC, and requires less computation time than FCA.

Table 12: Performance comparison for $k_* = 10$ on CENSUS dataset.

|  | Cost ($\downarrow$) | $\Delta$ ($\downarrow$) | Bal ($\uparrow$) | Computation time (sec) |
|---|---|---|---|---|
| SFC | 24.488 | 0.000 | 0.933 | 2753.30 |
| FCA | 11.804 | 0.000 | 0.934 | 3957.83 |
| FBC ✓ | 12.854 | 0.000 | 0.930 | 2927.42 |

**Unknown $K$**  We further analyze the posterior density of $K$, to validate the inferred $K$ is reasonable. Figure 13 shows the results, indicating that the inferred $K$ is sampled from the posterior mode, and posterior distributions are well-concentrated around the modes, similar to Figure 9.

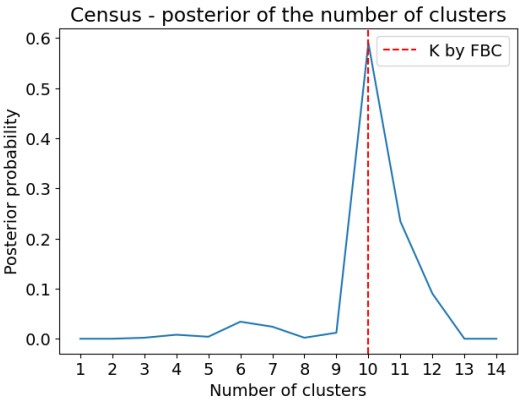

Figure 13: Posterior of $K$ on CENSUS dataset.

