# OpenReview forum: "Fair Bayesian Model-Based Clustering"
_ICLR.cc/2026/Conference — Submitted to ICLR 2026_

### Official Review · Reviewer_7XLq · 2025-10-29

**Soundness:** 3
**Presentation:** 3
**Contribution:** 3
**Rating:** 8
**Confidence:** 3

**Summary:**

The paper introduces a fair clustering algorithm based on a mixture model, namely FBC, which can automatically infer the number of clusters. In addition, an MCMC inference algorithm is proposed to obtain fair clustering results. The convergence of the algorithm is evaluated experimentally. Experimental results on three real-world datasets and three baseline methods demonstrate that the proposed method outperforms existing approaches.

**Strengths:**

1. The proposed fair clustering method based on a mixture model is novel and clearly presented.
2. The experimental results demonstrate the superior of the proposed fair clustering method in comparison with 3 well-known fair clustering models.

**Weaknesses:**

1. The paper does not explain the method to find the fixed number of cluster k* (in section 5.1).
2. The paper does not analyze the complexity (in big O notation) of the proposed MCMC algorithm (section 4).
3. The experiments are conducted on only several selected attributes of the datasets which might significantly affect to the results. Furthermore, the paper does not explain how those attributes are selected for experiments.

**Questions:**

See the weaknesses.
A minor comment: The paper states that “SFC is the first fair clustering method based on fairlets” (SFC (Backurs et al., 2019)). But this is not correct. The first fair clustering method based on fairlets is Chierichetti et al., (2017). Backurs et al., 2019 is an extension with “Scalable fair clustering”.

---

> ### Author Response · Authors · 2025-11-21
> **Response #1**
>
> We appreciate your great feedback and thoughtful remarks, and we have responded to all of your concerns as thoroughly as possible.
> In the revised paper, we marked the differences (e.g., new experimental results) in blue.
> Please note that the figure and section numbers mentioned in the comments correspond to those in the **revised version** of the paper.
>
> # Summary of changes
> * Main body:
>     - Section 1: We revised the introduction to place greater emphasis on the high-level contributions of our proposed fair Bayesian framework.
>     - Section 2: We added the explanations regarding an explanation with fairlets and the new proposition.
>     - Section 4: We moved the pseudo-code that was previously in Appendix into this section and noted that a corresponding visualization has been added to Appendix. We also included an explanation of how FBC can be interpreted from the objective–constraint perspective used in existing methods.
>
> * Appendix:
>     - Section A.2: We added an additional proposition (Proposition A.3) to analyze the maximum violation on the fairness measure $\Delta$, in the case of $r>0$. We added the discussion about this proposition in Section 2.1.2.
>     - Section B.1: We added an additional flow diagram to help understanding of FBC. We added a detailed explanation of FBC, in view of fairlets.
>     - Section F.1: We performed an additional experiment on the assignments for new observations, and report corresponding results.
>     - Section F.2: We further discuss about the scalability of the proposed algorithm, FBC.
>     - Section F.3: We performed an additional experiment on the new large-scale dataset, **Census**, and report corresponding results.
>
> * Minor: We highlighted the tables to improve the visibility of the results and corrected several typos. Also, we additionally inserted several sentences throughout the paper that may help improve understanding.

---

> ### Author Response · Authors · 2025-11-21
> **Response #2**
>
> # Responses
>
> > W1: The paper does not explain the method to find the fixed number of cluster k* (in section 5.1).
>
> - We consider the fixed $k_{ * }$ to compare FBC with other fair clustering algorithms that require a predefined number of clusters.
> Thus, whenever $k_{ * }$ is mentioned, we assume that it is given apriori. Note that FBC does not require the knowledge of $k_{ * }.$
>
> - As described in Section 5.1, it can be done by setting the prior for $K$ to be a Dirac delta at a fixed $k_{ * }.$ Additional clarification on this point is provided in Section B.3 in Appendix, in the paragraph titled `The case when $K$ is known'.
>
> > W2: The paper does not analyze the complexity (in big O notation) of the proposed MCMC algorithm (section 4).
>
> - We further discuss about the computational cost in Section F.2.
>  The computational complexity of the MH step (STEP 1) for $\mathbf{T}$ does not depend on the input dimension, and its scale on each iteration is $O(1)$ since it is a random swap with fixed time of repetition (10 times in our experiments).
>  The computational complexity of the mixture parameter sampling of each iteration (STEP 2) for $\Phi$, is empirically observed as $O(n)$ in [1].
>
> - Additionally, we provide empirical evidence by comparing its computation time with that of FCA (a baseline with similar performance), as reported in Table 2 in Section E.3.1.
>  As mentioned in Section 5.1, this difference primarily arises because FCA solves a linear program at every iteration to obtain $\mathbf{T}$, whereas our method searches for $\mathbf{T}$ stochastically.
>  Comparison with other baseline methods are also given in Section F.2.
>
>    [1] Mixture models with a prior on the number of components. Miller and Harrison. JASA. 2018.
>
> > W3: The experiments are conducted on only several selected attributes of the datasets which might significantly affect to the results. Furthermore, the paper does not explain how those attributes are selected for experiments.
>
> - For main experiments (Sections 5.1 and 5.2), across all datasets (Adult, Bank, Diabetes), **we adopt the same feature selection (continuous features)** used in prior fair clustering methods [2,3,4,5] to ensure fair comparisons.
>  Please note that the details about the selected attributes are given in Section E.1.
>
>  - For example:
>
>    Adult: we follow the feature selection of prior fair clustering methods [2,3,4].
>    That is, we use five continuous variables (age, fnlwgt, education-num, capital-gain, hours-per-week), gender as sensitive variable, and for the additional experiment with categorical variables, we follow FairDen [5] for the choice of 2 categorical features (marital status and race).
>
>    Bank: we follow the feature selection of prior fair clustering methods [3,4]. That is, we use six numerical variables (age, call duration, 3-month Euribor rate, number of employees, consumer price index, number of contacts during the campaign), for binary sensitive case, we set marital status as the sensitive attribute and grouping two categories (single, married) after excluding all unknown entries (following [4]), and for three category-sensitive attribute case, we divide marital status into three categories (single, married, divorced) excluding all unknown entries (following [3]).
>
>    [2] Scalable fair clustering. Backurs et al. ICML. 2019.
>
>    [3] Variational fair clustering. Ziko et al. AAAI. 2021.
>
>    [4] Fair clustering via alignment. Kim et al. ICML. 2025.
>
>    [5] Fairden: Fair density-based clustering. Krieger et al. ICLR. 2025.
>
> > Q1: A minor comment: The paper states that “SFC is the first fair clustering method based on fairlets” (SFC (Backurs et al., 2019)). But this is not correct. The first fair clustering method based on fairlets is Chierichetti et al., (2017). Backurs et al., 2019 is an extension with “Scalable fair clustering”.
>
> - That is correct. We were aware of this as well, but the statement was inaccurate. Thank you for pointing it out - we corrected it in the revision.

---

### Official Review · Reviewer_V2Dd · 2025-11-03

**Soundness:** 2
**Presentation:** 2
**Contribution:** 2
**Rating:** 4
**Confidence:** 2

**Summary:**

This paper proposes Fair Bayesian Clustering (FBC), a model-based approach for fair clustering that builds fairness directly into a Bayesian mixture model via a fairness prior whose mass concentrates on clusterings that respect proportional representation of sensitive groups in each cluster. Unlike most fair K-means variants, FBC can infer the number of clusters, works with data where a metric is hard to specify, and supports cluster-size constraints. The authors develop an MCMC inference scheme for the fair posterior and show on real datasets that FBC is competitive when K is known and can recover a “fair” K when K is unknown.

**Strengths:**

This paper introduces a Bayesian approach to fair clustering with a fair prior that concentrates posterior mass on proportionally fair clusterings, distinct from the K-means algorithms that dominate prior fair clustering work. This framing lets fairness be enforced probabilistically rather than via hard constraints.

The paper provides a concrete model specification and an MCMC inference scheme to approximate the posterior over the number of clusters, centers, and assignments under the fairness prior.

**Weaknesses:**

The paper does not clearly differentiate its Bayesian formulation from recent strong non–K-means fair clustering methods such as Fair Clustering via Alignment (FCA) [1] and FairDen [2], which already achieve strong fairness–utility trade-offs through alignment-based and density-based principles.

The proposed method, FBC. enforces only a proportional representation constraint across sensitive groups, which limits its flexibility in modeling fairness definitions that reflect real-world diversity, such as bounded or multi-attribute fairness. However, existing methods like FairKM [3] explicitly address multi-attribute fairness and could serve as benchmarks for evaluating FBC’s generality.

The MCMC-based inference proposed for FBC may not scale effectively to large or high-dimensional datasets, limiting its practicality relative to lighter fair clustering methods.

[1] Kim, Kunwoong, et al. "Fair Clustering via Alignment." arXiv preprint arXiv:2505.09131 (2025).

[2] Krieger, Lena, et al. "FairDen: Fair Density-Based Clustering." The Thirteenth International Conference on Learning Representations. 2025.

[1] Abraham, Savitha Sam, and Sowmya S. Sundaram. "Fairness in clustering with multiple sensitive attributes." arXiv preprint arXiv:1910.05113 (2019).

**Questions:**

Please refer to the weaknesses.

**Details Of Ethics Concerns:**

I do not identify any significant ethics concerns.

---

> ### Author Response · Authors · 2025-11-21
> **Response #1**
>
> Thank you for the detailed feedback and thoughtful inquiries. We have carefully responded to each of the issues you noted. In the revised paper, we marked the differences (e.g., new experimental results) in blue. Please note that the figure and section numbers mentioned in the comments correspond to those in the **revised version** of the paper.
>
> # Summary of changes
> * Main body:
>     - Section 1: We revised the introduction to place greater emphasis on the high-level contributions of our proposed fair Bayesian framework.
>     - Section 2: We added the explanations regarding an explanation with fairlets and the new proposition.
>     - Section 4: We moved the pseudo-code that was previously in Appendix into this section and noted that a corresponding visualization has been added to Appendix. We also included an explanation of how FBC can be interpreted from the objective–constraint perspective used in existing methods.
>
> * Appendix:
>     - Section A.2: We added an additional proposition (Proposition A.3) to analyze the maximum violation on the fairness measure $\Delta$, in the case of $r>0$. We added the discussion about this proposition in Section 2.1.2.
>     - Section B.1: We added an additional flow diagram to help understanding of FBC. We added a detailed explanation of FBC, in view of fairlets.
>     - Section F.1: We performed an additional experiment on the assignments for new observations, and report corresponding results.
>     - Section F.2: We further discuss about the scalability of the proposed algorithm, FBC.
>     - Section F.3: We performed an additional experiment on the new large-scale dataset, **Census**, and report corresponding results.
>
> * Minor: We highlighted the tables to improve the visibility of the results and corrected several typos. Also, we additionally inserted several sentences throughout the paper that may help improve understanding.

---

> ### Author Response · Authors · 2025-11-21
> **Response #2**
>
> ---
>
> > W1. The paper does not clearly differentiate its Bayesian formulation from recent strong non–K-means fair clustering methods such as...
>
> * **(High-level contribution of our fair Bayesian framework)** We would like to emphasize that, to the best of our knowledge, this is *8the first work to bring a Bayesian approach to the fair clustering problem.** While Bayesian clustering itself is a well-studied topic, it is not straightforward to adapt existing Bayesian clustering methods so that they satisfy fairness constraints.
>
>     In a Bayesian framework, the goal is to explore high-posterior regions of the parameter space. In the context of fair clustering, however, the posterior should be explored only within the fair region. This in turn requires the prior mass to be concentrated on the fair space, rather than on the entire unconstrained space. A naive way is to put fairness constraints on the parameter space, but this naive idea would not work well since the likelihood value is usually too small on the fairness constraint space and so exploring the posterior distribution on the constraint parameter space is numerically extremely difficult. We avoid this problem by introducing the matching function. Our Bayesian model does not have any constraint and so standard MCMC algorithm can be used without much hamper. To sum up, our matching-based formulation in Section 2 is, to our knowledge, **the first to explicitly construct such a prior restricted to the fair space but standard MCMC algorithm can be implemented easily.**
>
>     Moreover, from an optimization perspective, our formulation can be viewed as an analogue of a cost–fairness decomposition: **(i) the likelihood plays the role of the clustering cost, and (ii) the fair prior space plays a role similar to that of the fairness constraint.** In other words, we carefully design the fair-constrained space via the prior, while our MCMC algorithm stochastically searches over both the parameters and the matching functions within this fair-constrained space. Of course, our fair Bayesian model inherits most of advantages of standard Bayesian clustering methods offer crucial benefits such as ability to infer the number of clusters, way of dealing with non-numeric data (as long as the likelihood is defined) and ease to incorporate side conditions (e.g., satisfying upper bound of cluster size).
>
> - **(Advantage of our Bayesian model-based approach)**
>  We further explain the advantages of our Bayesian model-based clustering approach.
>  In cluster analysis, inferring an unknown number of clusters is itself a very important problem [1-5].
>  One advantage of our method is that it makes this possible in the context of fair clustering. Also, we aimed to bring additional advantages alongside inference on the number of clusters, which we have described in Section 5.3. The advantages are summarized as follows.
>
>    - **(Advantages of model-based clustering)**
>     First, as FBC is a model-based clustering method, it can be applied to a various data types as long as the likelihood is well-defined (e.g., categorical data, see Section E.3.3).
>     Furthermore, it can be used for density estimation (Section 5.2).
>    - **(Advantages of Bayesian clustering)**
>     Second, we can perform clustering under the constraint that the cluster sizes must be upper-bounded (Section E.3.4).
>     In contrast, to the best of our knowledge, only a post-processing method (i.e., [6]) handles this problem, which can disregard the underlying data distribution.
>     Our approach can naturally handle this upper-bound constraint itself and estimate the data density.
>
> - **(Comparison to FCA and FairDen)**
>  As described in Section 5.1, our method and FCA share the same underlying idea of matching, but the way they obtain the matching differs: FCA solves a linear program at every iteration, whereas our method searches stochastically.
>  **Consequently, as shown in Table 2 of Section E.3.1, our approach requires less computation time than FCA.**
>
>    The empirical comparison with FairDen is presented in Section E.3.3. According to those results, **our method can be applied to larger datasets than FairDen and also demonstrates stronger empirical performance.**
>
> [1] On Bayesian analysis of mixtures with an unknown number of components (with discussion). Richardson and Green. JRSSSB. 1997.
>
> [2] Bayesian analysis of mixture models with an unknown number of components-an alternative to reversible jump methods. Stephens. AOS. 2000.
>
> [3] Bayesian finite mixtures with an unknown number of components: The allocation sampler. Nobile and Fearnside. Statistics and Computing. 2007.
>
> [4] Estimation for the number of components in a mixture model using stepwise split-and-merge em algorithm. Xian et al. Pattern Recognition Letters. 2004.
>
> [5] On the number of components in a gaussian mixture model. McLachlan and Rathnayake. WIREs DMKD. 2014.
>
> [6] Fair algorithms for clustering. Bera et al. NeurIPS. 2019.

---

> ### Author Response · Authors · 2025-11-21
> **Response #3**
>
> ---
> > W2: The proposed method, FBC. enforces only a proportional representation constraint across sensitive groups, which limits its flexibility in modeling fairness definitions that reflect real-world diversity, such as bounded or multi-attribute fairness. However, existing methods like FairKM explicitly address multi-attribute fairness and could serve as benchmarks for evaluating FBC’s generality.
>
> -  **(Method/analysis for multinary sensitive attribute)**
>  We have already provided an extension to the case where the sensitive attribute is multinary (i.e., multi-category for a single sensitive attribute) in Section C.
>  Section C fully describes the model, the MCMC procedure, the fairness measure, and the theoretical results for the multinary sensitive-attribute setting.
>  The corresponding empirical results are presented in Table 4 of Section E.3.1.
>
>    Furthermore, when multiple sensitive attributes are considered simultaneously, resulting in a number of intersectional subgroups, one can also treat the problem in a multinary manner.
>
> - **(A future direction)**
>  On the other hand, if the number of sensitive attributes is very large, this may lead to high computational complexity and/or data sparsity issues.
>  We view this as an interesting direction for future research, along with the case of continuous sensitive attributes, and we plan to pursue it in the near future.
>  Thank you again for this helpful suggestion.
>
>
> ---
>
>
> > W3: The MCMC-based inference proposed for FBC may not scale effectively to large or high-dimensional datasets, limiting its practicality relative to lighter fair clustering methods.
>
> - **(Comparison of scalability in terms of computation time)**
>  As explained in Section 5.1, we compared computation time with FCA, which shows similar performance to ours (Table 2 in Section E.3.1).
>
> - **(Practical technique for efficient MCMC)**
>  First, as described in Section 4 (STEP 1 explanation), we propose repeatedly performing STEP 1, the MH sampling of $(\mathbf{T}',\mathbf{T}\_0',E')$. In our experiments, we repeated Step 1 for 10 times.
>  Second, we already know that optimal transport provides reasonably strong performance [7,8].
>  For this reason, we utilize optimal transport to obtain a good initial point as noted in the paper (algorithms explanation in Section E.2).
>
>  [7] Fair clustering through fairlets. Chierichetti et al. NeurIPS. 2017.
>
>  [8] Fair clustering via alignment. Kim et al. ICML. 2025.

---

### Official Review · Reviewer_D9oP · 2025-11-03

**Soundness:** 3
**Presentation:** 3
**Contribution:** 3
**Rating:** 4
**Confidence:** 3

**Summary:**

The work presents fair bayesian clustering, a novel method to address the limitation in existing fair clustering literature where users must specify the number of clusters, $k$, ahead of time. The proposed finite mixture model with unknown number of concepts is used in tandem with a "fair prior" to enact group fairness in the traditional sense.
The core contribution is a reduction of the group fairness constrain on balanced datasets to the existence of a matching map between instances of sensitive groups.
To avoid computational complexity in model parameter enumeration, the method places on a prior on this matching map instead with fairness enforced by construction.

**Strengths:**

### Compelling Theoretical Model + Results
- The connection between perfect group fairness and a matching map is the paper's strongest point. This is a very interesting reformulation of the fairness constraint, though I have some questions / concerns on this connection (see the following sections).
- To my knowledge this if the first work to take the Bayesian non-parametric approach to group fairness, which is. a clear and important contribution.

### Experimental Results
- The model is demonstrated to be competitive with non-bayesian methods when $K$ is fixed or reasonably inferred. These simulations seem consistent with the above theoretical insights.

**Weaknesses:**

### Model Issues
- The authors note early in the paper that "most existing fair clustering methods are based on the K-means clustering and thus
require the distance between instances and the number of clusters to be given in advance," but the matching map and "energy" is directly a function of distances between matched points. It seems contradictory to re-introduce such a major dependency when the paper claims to be avoiding the limitations of $k$-means. If a reasonable distance is difficult to define, then this prior becomes just as ill-defined as a $k$-means objective. This seemingly undermines one of the paper's central motivations--please correct me if I am wrong on this.
- The space of matching maps is massive and while verification checking may be fast, it seems the methods noted in the paper (like swapping in MH) would likely get stuck in local optima. The authors note that for faster convergence, they need to partially initalize $\mathcal{T}$ using optimal transport, which suggests that they are aware of this serious limitation in the methodology.
- The theoretical results seemingly breakdown unless very strong assumptions are imposed when $r > 0$. To circumvent the issues here, the solution is seemingly a "heuristic modification" or a very strong cardinality assumption (Proposition A.2). Thus, seemingly the main breakthroughs of this formulation fail to hold in the more common/general settings encountered within fair clustering.
- The lack of extension to new data points (not in training data) is seemingly a major practical limitation. A new test point has not corresponding clustering assignment under the matching map, so the model-based clustering algorithm seems severely weakened for practical utility beyond the fixed dataset.

### Missing Literature
The authors cite very few of the fair clustering papers, making it difficult to situate this result in the current landscape of known results. Specifically, it's odd that the authors note the main problem of fair clustering as pre-specification of the number of clusters, but many existing lines of work circumvent this exact issue. For example, fair hierarchical clustering is a major object of study in fair clustering that does not require such pre-specification but is not mentioned in this work. I encourage the authors to incorporate more review of the literature (see for example https://www.fairclustering.com)

**Questions:**

- What, if any, are the convergence guarantees of his MCM step procedure on a high-dimensional combinatorial space?
- Why is the distance computation in energy not the same issue as the noted central issue in $k$-means based clusterings?
- Can you discuss the hurdles in working on imbalanced data and why this framing is still useful in spite of them?

---

> ### Author Response · Authors · 2025-11-21
> **Response #1**
>
> We are grateful for your attentive review and valuable questions. We have made our best effort to resolve all points you raised. In the revised paper, we marked the differences (e.g., new experimental results) in blue. Please note that the figure and section numbers mentioned in the comments correspond to those in the **revised version** of the paper.
>
> # Summary of changes
> * Main body:
>     - Section 1: We revised the introduction to place greater emphasis on the high-level contributions of our proposed fair Bayesian framework.
>     - Section 2: We added the explanations regarding an explanation with fairlets and the new proposition.
>     - Section 4: We moved the pseudo-code that was previously in Appendix into this section and noted that a corresponding visualization has been added to Appendix. We also included an explanation of how FBC can be interpreted from the objective–constraint perspective used in existing methods.
>
> * Appendix:
>     - Section A.2: We added an additional proposition (Proposition A.3) to analyze the maximum violation on the fairness measure $\Delta$, in the case of $r>0$. We added the discussion about this proposition in Section 2.1.2.
>     - Section B.1: We added an additional flow diagram to help understanding of FBC. We added a detailed explanation of FBC, in view of fairlets.
>     - Section F.1: We performed an additional experiment on the assignments for new observations, and report corresponding results.
>     - Section F.2: We further discuss about the scalability of the proposed algorithm, FBC.
>     - Section F.3: We performed an additional experiment on the new large-scale dataset, **Census**, and report corresponding results.
>
> * Minor: We highlighted the tables to improve the visibility of the results and corrected several typos. Also, we additionally inserted several sentences throughout the paper that may help improve understanding.
>
> ---
> # Responses
>
> **(High-level contribution of our fair Bayesian framework)**
>
> We would like to emphasize that, to the best of our knowledge, this is **the first work to bring a Bayesian approach to the fair clustering problem.** While Bayesian clustering itself is a well-studied topic, it is not straightforward to adapt existing Bayesian clustering methods so that they satisfy fairness constraints.
>
> In a Bayesian framework, the goal is to explore high-posterior regions of the parameter space. In the context of fair clustering, however, the posterior should be explored only within the fair region. This in turn requires the prior mass to be concentrated on the fair space, rather than on the entire unconstrained space. A naive way is to put fairness constraints on the parameter space, but this naive idea would not work well since the likelihood value is usually too small on the fairness constraint space and so exploring the posterior distribution on the constraint parameter space is numerically extremely difficult. We avoid this problem by introducing the matching function. Our Bayesian model does not have any constraint and so standard MCMC algorithm can be used without much hamper. To sum up, our matching-based formulation in Section 2 is, to our knowledge, **the first to explicitly construct such a prior restricted to the fair space but standard MCMC algorithm can be implemented easily.**
>
> Moreover, from an optimization perspective, our formulation can be viewed as an analogue of a cost–fairness decomposition: **(i) the likelihood plays the role of the clustering cost, and (ii) the fair prior space plays a role similar to that of the fairness constraint.** In other words, we carefully design the fair-constrained space via the prior, while our MCMC algorithm stochastically searches over both the parameters and the matching functions within this fair-constrained space. Of course, our fair Bayesian model inherits most of advantages of standard Bayesian clustering methods offer crucial benefits such as ability to infer the number of clusters, way of dealing with non-numeric data (as long as the likelihood is defined) and ease to incorporate side conditions (e.g., satisfying upper bound of cluster size).

---

> ### Author Response · Authors · 2025-11-21
> **Response #2**
>
> ---
>
> > W1. The authors note early in the paper that "most existing fair clustering methods are based on the K-means clustering...
>
> * Thank you for the great question. Our use of a distance-based prior was primarily motivated by a prior work [1], which also employed the distance-based energy term in bipartite matching problem. However, our framework does not heavily rely on this choice, because **(i) the influence of the prior diminishes as the number of observations increases, and (ii) our empirical results show that its practical effect is indeed negligible} (see the detailed explanation below.**
>
>     Therefore, a random prior (or any prior that does not depend on distances) can also be used in our framework without materially affecting the results, and we are willing to revise the manuscript to clarify or de-emphasize the distance-based prior if the reviewer prefers.
>
>     Hence, we can say that using a random prior (or any prior not depending on the distance) in our proposed framework would be good, and we are willing to revise this point if you suggest.
>
> * **(Small impact of prior)** In standard Bayesian inference, as the number of observations increases, the influence of the prior term gradually diminishes and eventually vanishes.
>
>     For example, when $n$ is small, the search space of $\mathbf{T}$ is also small, so the choice of the initial $\mathbf{T}$ or the prior on $\mathbf{T}$ does not substantially affect performance. For general $n$, we may instead use a pre-specified initialization and prior to promote efficient mixing of $\mathbf{T}$ in the MCMC algorithm and thereby achieve faster convergence.
>
> * **(Empirical evidence)** Furthermore, an empirical evidence is already provided in Table 8 of Section E.3.5 in our paper. There, we empirically confirm that, once the number of observations is sufficiently large, changing the prior term does not affect the clustering performance.
>
> [1] Efficient sampling for bipartite matching problems. Volkovs and Zemel. NeurIPS. 2012.
>
> ---
>
> > W2. The space of matching maps is massive and while verification checking may be fast,...
>
> * **(Why finding local optima is reasonable)** Thank you for the excellent question. Because the space of matching maps is massive, we do not attempt to search for a good $\mathbf{T}$ over the entire space. Indeed, doing so is practically impossible in general (e.g., a large $n$). Instead, **our goal is to identify a local optimum, exploring at least the region where the posterior is high.** The finding of such local optima is empirically supported by Figure 10 in Section E.4, where `NLL` is decreased through time steps.
>
>     Additionally, **in MCMC algorithms that explore a large space, it is already common practice to use a local stochastic search rather than attempting to explore the entire space**. A well-known example of this strategy is Bayesian CART [2,3].
>
> * **(Practical techniques for efficient MCMC)** Technically, we employed a two-step strategy to address this issue. First, as described in Section 4, we propose repeatedly performing Step 1, the MH sampling of $(\mathbf{T}',\mathbf{T}_0',E')$. In our experiments, we repeated Step 1 for 10 times, as noted. Second, our use of an optimal transport map as the initial $\mathbf{T}$ is motivated by prior work showing that matching nearby instances tends to yield higher clustering utility (e.g., fairlet-based methods [4,5] and alignment-based methods [6]) (we noted this in the implementation details of Section E.2).
>
> [2] A Bayesian CART algorithm. Denison et al. Biometrika. 1998.
>
> [3] Bayesian CART Model Search. Chipman et al. JASA. 1998.
>
> [4] Fair clustering through fairlets. Chierichetti et al. NeurIPS. 2017.
>
> [5] Scalable fair clustering. Backurs et al. ICML. 2019.
>
> [6] Fair clustering via alignment. Kim et al. ICML. 2025.
>
> ---
>
> > W3. The theoretical results seemingly breakdown unless very strong assumptions are imposed...
>
> * **(Fairness violation when $r > 0$ and Proposition A.2)** We provide an additional theoretical result about the worst-case fairness violation (without the assumption in Proposition A.2), for the case of $n_1=\beta n_0+r$, in **Proposition A.3 in Section A.2**. The theoretical upper bound of $\Delta(Z)$ is $r(n_0-r)/n_0n_1,$ which is attained by the extreme case where all instances in $R_{\mathbf{T}}$ are assigned to the same cluster. This quantity can be further upper bounded as $r(n_0-r)/n_0n_1 \le 1 / ( \sqrt{\beta + 1} + \sqrt{\beta} )^{2},$ which is smaller when $\beta$ becomes larger.
>
>     The message of Proposition A.2 is that $\Delta(Z)$ becomes exactly zero when $|C_{k}^{(0)}|/n_0 = |C_{k}^{(0)}\cap R|/r.$ Also, note that we empirically verified that the fairness level is nearly perfect (e.g., Table 1) in experiments on benchmark datasets.

---

> ### Author Response · Authors · 2025-11-21
> **Response #3**
>
> ---
> > W4. The lack of extension to new data points (not in training data) is seemingly a major practical limitation. A new test point has not corresponding clustering assignment under the matching map, so the model-based clustering algorithm seems severely weakened for practical utility beyond the fixed dataset.
>
> * First, we would like to note that the previous methods we considered also did not address how to assign new data points [5,6,7].
>
> * As mentioned in the conclusion part, we considered this to be beyond the scope of our work, and therefore did not include it in the current manuscript. However, we do have a heuristic idea for extending the method, as follows.
>
>     Suppose the new dataset is given as $D_0^\mathrm{new}\cup D_1^\mathrm{new}.$ Then, we assign elements of $D_0^{\mathrm{new}}$ to their nearest neighbors in $D_0$, and similarly assign elements of $D_1^{\mathrm{new}}$ to their nearest neighbors in $D_1$. We empirically validate this assignment strategy as follows:
>     (i) we split the **Adult** dataset into training and test sets with an 8:2 ratio;
>     (ii) we perform FBC on the training data and then assign the test data according to the procedure described above.
>     The results are reported in Table 11 in Section F.1 of Appendix, where we observe that fairness on the test data (in terms of $\Delta$ and Balance) remains near the best values, and the NLD value is similar to that in Table 1, where the full dataset is used. These empirical findings suggest that the proposed assignment technique for new (test) data performs well in practice.
>
> * We are also interested in extending this idea so that it can be integrated more naturally into our proposed method, and we plan to pursue this direction as future work. In particular, extending the framework to an online setting, where new data can be incorporated by matching them to existing data via a suitable matching function, appears to be a promising direction that naturally follows from our current work.
>
> [7] Variational fair clustering. Ziko et al. AAAI. 2021.
>
> ---
>
> > W5. The authors cite very few of the fair clustering papers, making it difficult to situate this result in the current landscape of known results. Specifically, it's odd that the authors note the main problem of fair clustering as pre-specification of the number of clusters, but many existing lines of work circumvent this exact issue. For example, fair hierarchical clustering is a major object of study in fair clustering that does not require such pre-specification but is not mentioned in this work. I encourage the authors to incorporate more review of the literature (see for example https://www.fairclustering.com)
>
> * Thank you for sharing the materials along with your detailed explanation. As you noted, using hierarchical clustering does allow one to control the number of clusters.
>
>     However, choosing the final $K$ in this setting is a separate problem. Simply minimizing the `Cost` would push the solution toward selecting the largest possible $K$. Our approach differs conceptually in that we do not rely on heuristic choices for $K$; instead, we infer it in a scientifically grounded manner based on the posterior density.
>
> ---
>
> > Q1. What, if any, are the convergence guarantees of this MCMC step procedure on a high-dimensional combinatorial space?
>
> * We demonstrated empirical convergence in Section E.4. Also, please refer to our response to Weakness-2.
>
> ---
>
> > Q2. Why is the distance computation in energy not the same issue as the noted central issue in $k$-means based clusterings?
>
> * Please see our response to Weakness 1.
>
> ---
>
> > Q3. Can you discuss the hurdles in working on imbalanced data and why this framing is still useful in spite of them?
>
> * We would like to emphasize a key strength of our method that, it is capable of achieving the highest level of fairness even under extreme imbalance. For example, suppose we have a single instance from $\mathcal{D}^{(0)}$ and 100 instances from $\mathcal{D}^{(1)}$. Because all instances are grouped through the matching function $\mathbf{T}$, the procedure produces a single cluster containing all instances. This outcome illustrates the advantage of incorporating a matching function.

---

### Official Review · Reviewer_4Lv3 · 2025-11-06

**Soundness:** 2
**Presentation:** 2
**Contribution:** 3
**Rating:** 6
**Confidence:** 3

**Summary:**

This paper addresses significant limitations in the field of fair clustering, where most existing methods are derivatives of K-means. These methods typically require (1) the number of clusters ($k$) to be specified in advance and (2) a pre-defined distance metric, rendering them inflexible for datasets with an unknown cluster structure or non-trivial data types (e.g., categorical data). To solve this, the authors propose a "Fair Bayesian Clustering" (FBC) model.  The core technical innovation is a "fair prior", constructed to put its mass only on fair clusters, ensuring that the model's posterior distribution naturally favors solutions that satisfy group fairness with moderate violations. They develop an efficient MCMC algorithm for posterior approximation. Experimental evidence on an extend number of datasets show the method to beat most benchlines, achieving high fairness with minimal loss in performance. Especially, the algorithm's performance matches that of the fixed-number flat clustering algorithms in the best case and infers the number of clusters correctly, showing it to be effective.

**Strengths:**

The paper tackles a well-known and practical shortcoming of the dominant K-means paradigm in fair clustering. The inflexibility of fixed $k$ is a major hurdle in real-world applications. The move to a Bayesian mixture model is also a natural and powerful way to solve these problems. The ability to infer $k$ is a significant advantage for exploratory analysis, and the model-based framework is inherently more adaptable to diverse data types than distance-based algorithms.

The idea of a "fair prior" is very appealing. Instead of enforcing fairness as a hard constraint during optimization (which can be computationally complex) or as a post-processing step (which can be suboptimal), this approach bakes fairness directly into the model's assumptions. If the prior is well-designed, the MCMC sampler will naturally explore a "fair" solution space.

The authors explicitly acknowledge the difficulty of MCMC on constrained spaces and present their "matching" prior as a solution. This focus on developing a novel prior that also leads to an efficient MCMC algorithm is a key strength.

Finally, the empirical evidence is strong in the sense that the inferred number of clusters is mostly correct, and it achieves comparable performance with the previous methods in the best scenario, with even less running time.

**Weaknesses:**

Despite the earlier claim that the paper focuses on the "perfect fairness" scenario, in section 2.1.2., when $n_1 = \beta n_0 + r$, the fairness constraint can actually be moderately violated. I think the constraint violation should be further quantified in the main body.

Although the Bayesian fair prior and posterior update framework is new, the idea of "coupling" points from different sensitivity groups follows the tradition in the fair clustering community, but is more difficult to follow in the presentation of this paper. This paper introduces the mapping of points in the larger sensitivity group to the smaller, but then they evolved to messy notations like $T, T_0, E$ etc. Also in general the notation system can benefit from some clean-ups.

**Questions:**

Can you elaborate more on the case where $n_1 = \beta n_0 + r$ and why fairness is not guaranteed when you pick functions from the set of mappings $\mathcal{T}$? I think this is an important point to clarify and it'll be better if you can give concrete examples. Is the problem that the cluster assignment might not put the mapped sets of different sizes proportionally into different clusters? If so, can't you incorporate that constraint into the model?

I'm wondering if you can phrase the problem and solution differently using the language of fairlets as introduced in previous papers. The mapping set $\mathcal{T}$ that you are constructing seems to be equivalent to the definition of all possible decomposition of fairlets. It would be easier to understand in that sense.

I didn't find discussion on how well the new method scales compared to other methods as dataset size increases. Any ideas on that?

---

> ### Author Response · Authors · 2025-11-21
> **Response #1**
>
> We appreciate your thorough reviews and insightful questions, and we have worked hard to address all of your comments and concerns. In the revised paper, we marked the differences (e.g., new experimental results) in blue. Please note that the figure and section numbers mentioned in the comments correspond to those in the **revised version** of the paper.
>
> # Summary of changes
> * Main body:
>     - Section 1: We revised the introduction to place greater emphasis on the high-level contributions of our proposed fair Bayesian framework.
>     - Section 2: We added the explanations regarding an explanation with fairlets and the new proposition.
>     - Section 4: We moved the pseudo-code that was previously in Appendix into this section and noted that a corresponding visualization has been added to Appendix. We also included an explanation of how FBC can be interpreted from the objective–constraint perspective used in existing methods.
>
> * Appendix:
>     - Section A.2: We added an additional proposition (Proposition A.3) to analyze the maximum violation on the fairness measure $\Delta$, in the case of $r>0$. We added the discussion about this proposition in Section 2.1.2.
>     - Section B.1: We added an additional flow diagram to help understanding of FBC. We added a detailed explanation of FBC, in view of fairlets.
>     - Section F.1: We performed an additional experiment on the assignments for new observations, and report corresponding results.
>     - Section F.2: We further discuss about the scalability of the proposed algorithm, FBC.
>     - Section F.3: We performed an additional experiment on the new large-scale dataset, **Census**, and report corresponding results.
>
> * Minor: We highlighted the tables to improve the visibility of the results and corrected several typos. Also, we additionally inserted several sentences throughout the paper that may help improve understanding.
>
> ---
> # Responses
>
> > W1. Despite the earlier claim that the paper focuses on the "perfect fairness" scenario, in section 2.1.2., when $n_1=\beta n_0+r$, the fairness constraint can actually be moderately violated. I think the constraint violation should be further quantified in the main body.
>
> * We would like to clarify that, **in such cases, other methods also cannot achieve perfect fairness**, when using hard assignments. We introduced separate cases to structure the modeling, and for the case ($r>0$), although perfect fairness is theoretically unattainable, our approach is designed heuristically to achieve performance as close to it as possible.
>
> * In this kind of situation, we rather focus on identifying specific mappings $\mathbf{T}$ that can provide fair assignments ($\mathcal{T}_R$), and we design the proposal for $\mathbf{T}$ so that the proposal stays in the space $\mathcal{T}_R$. The effectiveness of our design can be seen in the empirical evaluation. As shown in Table 1, FBC successfully minimizes the fairness measure $\Delta$.
>
> * **(Worst-case fairness violation when $r > 0$)** We provide an additional theoretical result about the worst-case fairness violation, for the case of $n_1=\beta n_0+r$, in **Proposition A.3 in Section A.2**. The theoretical upper bound of $\Delta(Z)$ is $r(n_0-r)/n_0n_1,$ which is attained by the extreme case where all instances in $R_{\mathbf{T}}$ are assigned to the same cluster. This quantity can be further upper bounded as $r(n_0-r)/n_0n_1 \le 1 / ( \sqrt{\beta + 1} + \sqrt{\beta} )^{2},$ which is smaller when $\beta$ becomes larger. $\Delta(Z)$ becomes exactly zero when $|C_{k}^{(0)}|/n_0 = |C_{k}^{(0)}\cap R|/r$ (Proposition A.2). Also, note that we empirically verified that the fairness level is nearly perfect (e.g., Table 1) in experiments on benchmark datasets.
>
> ---
>
> > W2. Although the Bayesian fair prior and posterior update framework is new, the idea of "coupling" points from different sensitivity groups follows the tradition in the fair clustering community, but is more difficult to follow in the presentation of this paper. This paper introduces the mapping of points in the larger sensitivity group to the smaller, but then they evolved to messy notations like $T,T_0,E$ etc. Also in general the notation system can benefit from some clean-ups.
>
> * Basically, when $n_1$ and $n_0$ are the same or an integer multiple and we pursue a perfectly fair clusters, messy notations such as $T_0$ and $E$ are not necessary. We introduce such messy notations for the completeness of our proposed algorithm to able to process unbalanced data under positive fairness gap.
>
> * To improve clarity, we added a new flow diagram (Figure 6 in Section B.1) and moved the pseudo-code into the main body (Algorithm 1 in Section 4).

---

> ### Author Response · Authors · 2025-11-21
> **Response #2**
>
> ---
> > Q1. Can you elaborate more on the case where $n_1=\beta n_0+r$ and why fairness is not guaranteed when you pick functions from the set of mappings $\mathcal{T}$? I think this is an important point to clarify and it'll be better if you can give concrete examples. Is the problem that the cluster assignment might not put the mapped sets of different sizes proportionally into different clusters? If so, can't you incorporate that constraint into the model?
>
> * A detailed explanation of this can be found in Section A.2.
> * Please also refer to our response to Weakness-1.
>
> ---
>
> > Q2. I'm wondering if you can phrase the problem and solution differently using the language of fairlets as introduced in previous papers. The mapping set $\mathcal{T}$ that you are constructing seems to be equivalent to the definition of all possible decomposition of fairlets. It would be easier to understand in that sense.
>
> * Thank you very much for the constructive comment and we agree that it would be easier to understand our framework if explained by the language of fairlets. Hence, we added the following explanation in the revision (Sections 2.1.2 and B.1):
>
>     For simplicity, let ignore $\mathbf{T}_0$ and $E$, and consider only the perfect-fairness case with $\mathbf{T}$. In this setting, **$\mathbf{T}$ can be seen as a map to build fairlets, and we treat $\mathbf{T}$ as a random variable to perform posterior inference.** Inspired by MFM [1] which obtains MCMC samples of model parameters, we integrate the MH algorithm into the original MCMC procedure to yield posterior samples of $\mathbf{T}$. We also theoretically validate this proposed algorithm in Section B.3. Moreover, for reasonable initialization and prior for $\mathbf{T}$, we incorporate ideas from optimal transport inspired of the the fairlet-based approaches. A crucial difference is that, we update $\mathbf{T}$ for higher clustering utility (e.g., higher log-likelihood or lower clustering cost) rather than using a fixed $\mathbf{T},$ as fairlet-based approaches do.
>
> [1] Mixture models with a prior on the number of components. Miller and Harrison. JASA. 2018.
>
> ---
>
> > Q3. I didn't find discussion on how well the new method scales compared to other methods as dataset size increases. Any ideas on that?
>
> * We added a plot showing dataset size vs. computation time (Figure 12 in Section F.2 of Appendix). We can observe that FBC is competitive to baseline methods in terms of computation time, so we can conclude that FBC is scalable as other methods are.

---

### Official Review · Reviewer_SoZQ · 2025-11-07

**Soundness:** 2
**Presentation:** 2
**Contribution:** 2
**Rating:** 4
**Confidence:** 3

**Summary:**

This paper proposes Fair Bayesian Clustering (FBC), a Bayesian model-based clustering framework that incorporates group fairness into mixture models. The key idea is to design a fair prior over cluster assignments based on a matching map between samples from different sensitive groups, ensuring that matched samples are assigned to the same cluster. An efficient MCMC inference algorithm is developed to approximate the posterior distribution over the number of clusters, cluster parameters, and the fairness-related latent variables. The method allows automatic inference of the number of clusters, supports both continuous and categorical data, and can incorporate size constraints. Experiments on benchmark datasets show that FBC achieves comparable fairness–utility trade-offs compared with existing fair clustering algorithms, while being able to infer the number of clusters automatically.

**Strengths:**

1. The paper is among the first to integrate group fairness constraints into a Bayesian mixture modeling framework (an existing model for clustering without fairness constraints), providing new insights into fairness-aware clustering and Bayesian non-parameter.

2. The proposed  fair prior through the matching map is sound, which allows inference via standard Bayesian sampling tools without strict constraints on parameters.

3. Unlike prior fair clustering methods (which are typically combined with $k$-means objectives), the proposed FBC model can infer the number of clusters $k$ through a hierarchical prior, which is an interesting extension.

**Weaknesses:**

1. The matching based ideas have also been used in fairlet and fair $k$-means methods, and here it is mainly reformulated as a Bayesian prior combined with MCMC. Although the paper claims to be the first to apply this idea in a Bayesian clustering setting, the core mechanism remains very similar to previous approaches, making it difficult to identify clear Bayesian advantages beyond inferring the number of clusters.

2. This paper lacks a comprehensive comparison with previous fair clustering models. Integrating clustering objectives with fairness is natural as it can guarantee the fraction of data points from each protected group while minimizing the clustering costs to make the grouping (clustering) results convincable. However, the proposed model in this paper does not consider optimizing the clustering costs, where a detailed discussion should be included.

3. The proposed method does not include a principled way to assign new data points at test time, which the authors acknowledge as future work. For a clustering method intended for practical use, this omission is significant.

4. Without explicit pseudo-code or a flow diagram, it is difficult for readers to fully understand the operational steps and practical implementation of the algorithm.

5. A stated advantage of FBC is its capability to balance clustering utility and group fairness within a Bayesian framework. Nevertheless, the experiments only report individual metrics but do not present evidences that can highlight the trade-off between utility and fairness. Many tables do not bold or highlight the best results among the compared algorithms. This makes it difficult for readers to quickly identify performance differences and assess the relative strength of FBC.

**Questions:**

1. Can the authors clarify why inferring k is particularly important in fair clustering, and demonstrate cases where this leads to qualitatively different or better fairness outcomes compared to fixing k?

2. Can the authors provide a detailed discussion on the advantages of not simultaneously considering optimizing the clustering objectives in the proposed model?

3. Can the proposed FBC algorithm efficiently handle large-scale datasets such as KDDCUP99? Moreover, can FBC still infer a reasonable number of clusters K when applied to such large datasets?

4. How sensitive are the experimental results to the choice of hyperparameters?  Could different parameter settings significantly change the fairness–utility balance or the inferred number of clusters?

---

> ### Author Response · Authors · 2025-11-21
> **Response #1**
>
> Thank you for your careful reviews and thoughtful questions. We have done our best to address all of the points and concerns you raised. In the revised paper, we marked the differences (e.g., new experimental results) in blue. Please note that the figure and section numbers mentioned in the comments correspond to those in the **revised version** of the paper.
>
> # Summary of changes
> * Main body:
>     - Section 1: We revised the introduction to place greater emphasis on the high-level contributions of our proposed fair Bayesian framework.
>     - Section 2: We added the explanations regarding an explanation with fairlets and the new proposition.
>     - Section 4: We moved the pseudo-code that was previously in Appendix into this section and noted that a corresponding visualization has been added to Appendix. We also included an explanation of how FBC can be interpreted from the objective–constraint perspective used in existing methods.
>
> * Appendix:
>     - Section A.2: We added an additional proposition (Proposition A.3) to analyze the maximum violation on the fairness measure $\Delta$, in the case of $r>0$. We added the discussion about this proposition in Section 2.1.2.
>     - Section B.1: We added an additional flow diagram to help understanding of FBC. We added a detailed explanation of FBC, in view of fairlets.
>     - Section F.1: We performed an additional experiment on the assignments for new observations, and report corresponding results.
>     - Section F.2: We further discuss about the scalability of the proposed algorithm, FBC.
>     - Section F.3: We performed an additional experiment on the new large-scale dataset, **Census**, and report corresponding results.
>
> * Minor: We highlighted the tables to improve the visibility of the results and corrected several typos. Also, we additionally inserted several sentences throughout the paper that may help improve understanding.
> ---
> # Responses
>
> **(High-level contribution of our fair Bayesian framework)**
>
> We would like to emphasize that, to the best of our knowledge, this is **the first work to bring a Bayesian approach to the fair clustering problem.** While Bayesian clustering itself is a well-studied topic, it is not straightforward to adapt existing Bayesian clustering methods so that they satisfy fairness constraints.
>
> In a Bayesian framework, the goal is to explore high-posterior regions of the parameter space. In the context of fair clustering, however, the posterior should be explored only within the fair region. This in turn requires the prior mass to be concentrated on the fair space, rather than on the entire unconstrained space. A naive way is to put fairness constraints on the parameter space, but this naive idea would not work well since the likelihood value is usually too small on the fairness constraint space and so exploring the posterior distribution on the constraint parameter space is numerically extremely difficult. We avoid this problem by introducing the matching function. Our Bayesian model does not have any constraint and so standard MCMC algorithm can be used without much hamper. To sum up, our matching-based formulation in Section 2 is, to our knowledge, **the first to explicitly construct such a prior restricted to the fair space but standard MCMC algorithm can be implemented easily.**
>
> Moreover, from an optimization perspective, our formulation can be viewed as an analogue of a cost–fairness decomposition: **(i) the likelihood plays the role of the clustering cost, and (ii) the fair prior space plays a role similar to that of the fairness constraint.** In other words, we carefully design the fair-constrained space via the prior, while our MCMC algorithm stochastically searches over both the parameters and the matching functions within this fair-constrained space. Of course, our fair Bayesian model inherits most of advantages of standard Bayesian clustering methods offer crucial benefits such as ability to infer the number of clusters, way of dealing with non-numeric data (as long as the likelihood is defined) and ease to incorporate side conditions (e.g., satisfying upper bound of cluster size).

---

> ### Author Response · Authors · 2025-11-21
> **Response #2**
>
> ---
> > W1. The matching based ideas have also been used in fairlet and fair $k$-means methods, and here it is mainly reformulated as a Bayesian prior combined with MCMC. Although the paper claims to be the first to apply this idea in a Bayesian clustering setting, the core mechanism remains very similar to previous approaches, making it difficult to identify clear Bayesian advantages beyond inferring the number of clusters.
>
> * We agree that our method shares the matching-based mechanism. However, modifying existing fair $K$-means algorithms for Bayesian inference would not be easy. As we explained above, exploring the posteriors under the fairness constraint would be computationally very demanding. We avoid this problem by introducing the matching function to Bayesian models which allows us to use a standard MCMC algorithm to explore  the posteriors of fair models.
>
> * One may argue what are advantages of Bayesian clustering over $K$-means clustering. As we described in Sections 5.2 and 5.3, advantages of Bayesian model-based clustering beyond inferring $K$, are summarized as follows.
>
> * **(Advantages of model-based clustering)** First, as FBC is a model-based clustering method, it can be applied to a various data types as long as the likelihood is well-defined (e.g., categorical data, see Sections 5.3 and E.3.3). Furthermore, it can be used for density estimation (Section 5.2).
>
> * **(Advantages of Bayesian clustering)** Second, we can perform clustering under the constraint that the cluster sizes must be upper-bounded (Sections 5.3 and E.3.4). In contrast, to the best of our knowledge, only a post-processing method (i.e., [1]) handles this problem, which can disregard the underlying data distribution. Our approach can naturally handle this upper-bound constraint itself and estimate the data density.
>
> [1] Fair algorithms for clustering. Bera et al. NeurIPS. 2019.
>
> ---
>
> > W2. This paper lacks a comprehensive comparison with previous fair clustering models...
> * As described in equations (4–6) of Section 2, we consider a mixture model and consider a fairness constraint on the latent assignment variable $Z$. When perform posterior inference under our setup, getting a MAP solution corresponds to **optimizing the mixture model likelihood under the given constraint, which is closely related to the clustering cost.**
>
> * In particular, if we set the covariance of each mixture component to be the identity matrix, the resulting negative log-likelihood (NLL) becomes almost identical to the clustering cost used in previous fair clustering models. That is, if the assignment probability is confident (i.e., near 1 for one cluster and near 0 for others), then NLL is very close to the clustering cost. Thus, as a by-product, our posterior inference naturally leads to a reduction in this cost, and we demonstrated this effect through our empirical performance evaluation (Section 5).
>
> ---
>
> > W3. The proposed method does not include a principled way to assign new data points at test time...
>
> * First, we would like to note that the previous methods we considered also did not address how to assign new data points [2,3,4].
>
> * As mentioned in the conclusion part, we considered this to be beyond the scope of our work, and therefore did not include it in the current manuscript. However, we do have a heuristic idea for extending the method, as follows.
>
> * Suppose the new dataset is given as $D_0^\mathrm{new}\cup D_1^\mathrm{new}.$ Then, we assign elements of $D_0^{\mathrm{new}}$ to their nearest neighbors in $D_0$, and similarly assign elements of $D_1^{\mathrm{new}}$ to their nearest neighbors in $D_1$.
>
>     We empirically validate this assignment strategy as follows:
>     (i) we split the **Adult** dataset into training and test sets with an 8:2 ratio;
>     (ii) we perform FBC on the training data and then assign the test data according to the procedure described above.
>     The results are reported in Table 11 in Section F.1 of Appendix, where we observe that fairness levels on the test data (in terms of $\Delta$ and Balance) remains near the highest values, and the NLD value is similar to that in Table 1, where the full dataset is used for both training and testing.
>
> * We are also interested in extending this idea so that it can be integrated more naturally into our proposed method, and we plan to pursue this direction as future work. In particular, extending the framework to an online setting, where new data can be incorporated by matching them to existing data via a suitable matching function, appears to be a promising direction that naturally follows from our current work. These empirical findings suggest that the proposed assignment technique for new (test) data performs well in practice.
>
> [2] Scalable fair clustering. Backurs et al. ICML. 2019.
>
> [3] Variational fair clustering. Ziko et al. AAAI. 2021.
>
> [4] Fair clustering via alignment. Kim et al. ICML. 2025.

---

> ### Author Response · Authors · 2025-11-21
> **Response #3**
>
> ---
>
> > W4. Without explicit pseudo-code or a flow diagram, it is difficult for readers to fully understand the operational steps and practical implementation of the algorithm.
>
> * Due to the current page limit, the pseudo-code was placed in Appendix (Section B.3 in the old version). To improve clarity, we added a new flow diagram (Figure 6 in Section B.1 in Appendix) and moved the pseudo-code into the main body (Algorithm 1 in Section 4).
>
> ---
> > W5. A stated advantage of FBC is its capability to balance clustering utility and group fairness within a Bayesian framework. Nevertheless, the experiments only report individual metrics but do not present evidences that can highlight the trade-off between utility and fairness. Many tables do not bold or highlight the best results among the compared algorithms. This makes it difficult for readers to quickly identify performance differences and assess the relative strength of FBC.
>
> * To help readers’ understanding, we applied boldface to the best values and underlining to the second-best values for both `Cost` and $\Delta$ in the tables. Thank you for your suggestion.
>
> ---
>
> > Q1. Can the authors clarify why inferring k is particularly important in fair clustering,...
>
> * **(Why inferring $K$ matters in fair clustering)**
> If there is a prior knowledge about $K,$ of course we can use it.
> However, in many real-world applications, such prior information is not available, and so a way of inferring the number of clusters is a valuable tool.
> There is a vast amount of literature about the choice of the optimal number of clusters for standard clustering problems (without fairness constraints) [5,6,7,8,9], and Bayesian model-based clustering is known to be an efficient one for this purpose [10,11,12]. Our aim is to extend the standard Bayesian clustering algorithm to incorporate fairness constraints, so that most of advantages of Bayesian clustering remain intact.
>
>     For a given dataset, satisfying fairness under some values of $K$ may not be reasonable (e.g., due to very low utility).
>     For example: If $K$ is too small, the algorithm is forced to merge heterogeneous groups to satisfy fairness, which can obscure meaningful structure. If $K$ is too large, clusters become small and sparse, so the fairness constraints become harder to satisfy for small clusters and estimation may become unstable. Figure 1 in Section 1 is an example for this explanation. Thus, in a fair setting, the `right' $K$ is strongly data- and fairness-dependent and is generally difficult to determine in advance.
>
>     Our Bayesian framework (FBC) provides **one principled way of inferring $K$ under such fairness constraints.** Rather than fixing $K$ and running separate fair clustering procedures for $K=2,3,4,\dots$, we place a prior on $K$ and let the posterior concentrate on values of $K$ that **simultaneously yield (i) high likelihood, which reflects the underlying data geometry and density, and (ii) fair partitions under the constraints.** In this way, **FBC automatically infers a reasonable number of fair clusters driven by the data, without ad-hoc tuning of $K$.**
>
>     Empirically, we also observe that the $K$ inferred by FBC is reasonable from multiple perspectives, including cluster quality and density estimation (Section 5.2).
>
> [5] Estimating the number of clusters in a data set via the Gap Statistic. Tibshirani et al. JRSSSB. 2001.
>
> [6] A dendrite method for cluster analysis. Calinski and Harabasz. Communications in Statistics. 1974.
>
> [7] Silhouettes: A graphical aid to the interpretation and validation of cluster analysis. J. Comput. Appl. Math. 1987.
>
> [8] An examination of procedures for determining the number of clusters in a data set. Milligan and Cooper. Psychometrika. 1985.
>
> [9] Finding groups in data: an introduction to cluster analysis. Kaufman and Rousseeuw. 2009.
>
> [10] Markov Chain Sampling Methods for Dirichlet Process Mixture Models. Neal. J. Comput. Graph. Stat. 2000.
>
> [11] On bayesian analysis of mixtures with an unknown number of components (with discussion). Richardson and Green. JRSSSB. 1997.
>
> [12] Bayesian Analysis of Mixture Models with an Unknown Number of Components. Stephens. AOS. 2000.
>
> ---
>
> > Q2. Can the authors provide a detailed discussion on the advantages of not simultaneously considering optimizing the clustering objectives in the proposed model?
>
> * We may maximize the posterior to have the MAP (Maximum A Posteriori) estimator. Instead, we use an MCMC algorithm to explore high posterior regions of fair models. That is, we **use a stochastic search rather than a deterministic optimization.** **An obvious advantage of stochastic search is that we can find a good non-numeric parameter such as the number of clusters.** In addition, we can incorporate side conditions such as the upper bound of the size of each cluster (Section 5.3). Deterministic optimization would not be directly applicable for such problems.

---

> ### Author Response · Authors · 2025-11-21
> **Response #4**
>
> ---
>
> > Q3. Can the proposed FBC algorithm efficiently handle large-scale datasets such as KDDCUP99? Moreover, can FBC still infer a reasonable number of clusters K when applied to such large datasets?
>
> * Thank you for introducing the additional large-scale dataset. However, it appears that this dataset has not been used in the existing fairness literature, and we are not sure that arbitrarily designating a sensitive attribute from its features may not be ethically appropriate.
>
>     Instead, we conducted experiments using the **Census** dataset, which has a similar scale and has been previously used in the fairness literature. The results are provided in Table 12 of Section F.3, showing the similar behavior to that of Table 1 (the results for moderate-sized datasets). That is, while all methods achieves near-perfect fairness, FBC yields better clustering cost than SFC, and requires less computation time than FCA. Also, Figure 13 indicates that the inferred $K$ is sampled from the posterior mode, and posterior distributions are well-concentrated around the modes, similar to Figure 9 (the results for moderate-sized datasets).
>
> ---
>
> > Q4. How sensitive are the experimental results to the choice of hyperparameters? Could different parameter settings significantly change the fairness–utility balance or the inferred number of clusters?
>
> * **(Prior and $R$)** An ablation study on the prior and the choice of $R$ is provided in Section E.3.5. The results indicate that variations in these two components do not have a significant impact on performance.
>
> * **(For $\kappa$)** In [5], the prior work on finite mixture models, the prior on $K$ was set to $\mathrm{Geometric}(\kappa)$. However, this choice may lead to results that depend on the value of $\kappa$. As an additional safeguard, we additionally introduced a hierarchical prior on $\kappa$. Further details can be found in Section D.
>
> [5] Mixture models with a prior on the number of components. Miller and Harrison. JASA. 2018.

---

> ### Comment · Reviewer_SoZQ · 2025-11-26
> **Reponse to the Authors**
>
> Thank you for the detailed rebuttal and the revised manuscript. The added explanations of the Bayesian formulation, the role of the matching function, how K is inferred, and how new data points are handled, together with the new experiments (including fairness–utility trade-off visualizations and hyperparameter ablations), address several of my earlier concerns about clarity and empirical support. However, I still feel that the conceptual novelty over existing fair $k$-means and Bayesian mixture models is somewhat limited, and the experiments do not yet convincingly demonstrate clear advantages on truly large-scale (i.e., over 10M) or more challenging real-world fairness scenarios. Thus, I would like to maintain my original overall score.

---

> > ### Author Response · Authors · 2025-11-26
> > **Response**
> >
> > We would like to more clearly convey the overall flow and novelty of our work.
> >
> > Existing fair clustering methods are limited by their reliance on distance-based objectives and the requirement of pre-specified number of clusters $K.$
> > Introducing Bayesian mixture models could address these limitations.
> > However, injecting fairness constraints into Bayesian mixture models is non-trivial and not straightforward.
> >
> > To overcome this difficulty, we leverage the idea of matching (or fairlet) methods and reformulate the problem by restricting the prior to a fair space induced by this matching. Although the notion of matching itself is not new, our core technical contribution lies in treating the matching as a random variable and successfully integrating its posterior inference into a Bayesian framework, both theoretically and empirically. This allows our algorithm to maintain the strengths of Bayesian and model-based approaches (e.g., inference of $K$, applicability to categorical data) while offering new benefits (e.g., handling cluster size upper bounds).
> >
> > We believe that this methodological and technical development constitutes our main contribution. We would be very grateful if you could take these points into careful consideration. Thank you.

---

### Meta-Review · Area_Chair_QTE9 · 2026-01-05

**Summary:**

This paper proposes Fair Bayesian Clustering (FBC), a Bayesian mixture modeling framework that incorporates group fairness through a matching-based prior and MCMC inference. This method can infer the number of clusters automatically. Reviewers acknowledge the integration of fairness into a Bayesian clustering framework, and find the connection between group fairness and a matching map to be theoretically interesting. In the original review, reviewers note that the contribution over existing matching- and fairlet-based approaches is limited, and the Bayesian formulation offering limited additional advantages beyond inferring the number of clusters. Reviewers also raised concerns regarding the method’s advantage on large-scale datasets or more challenging real-world fairness settings. Furthermore, the experimental evaluation does not clearly demonstrate fairness–utility trade-offs, omits key comparisons to recent fair clustering methods, and lacks clarity in presentation and algorithm details. Though some concerns were addressed in the rebuttal, there are remaining concerns regarding limited novelty and insufficient empirical validation. Given these concerns, I recommend rejecting the paper in its current form.

**Reviewer Concerns:**

The rebuttal addressed concerns regarding fairness–utility trade-offs and algorithmic details, but issues of limited novelty and insufficient empirical validation remain.

**Reviewer Scores:**

Reviewer SoZQ explicitly mentioned in the response that they would keep their original score of 4 due to remaining concerns about novelty and empirical validation.

Reviewer 4Lv3, D9oP and V2Dd also raised concerns regarding the novelty of the work. I'd expect their final evaluations to be borderline or negative.

Reviewer 7XLq's original questions are mainly about clarification of algorithm details, complexity, experimental setting, which were addressed in the rebuttal. I'd expect their final score to remain positive.

---

### Decision · Program_Chairs · 2026-01-26

Reject